# Cryo-EM structures of the plant anion channel SLAC1 from *Arabidopsis thaliana* suggest a combined activation model

Yeongmok Lee [1], Hyeon Seong Jeong[2,3], Seoyeon Jung [1], Junmo Hwang[2], Chi Truc Han Le[1], Sung-Hoon Jun [4], Eun Jo Du[2], KyeongJin Kang [2], Beom-Gi Kim[5], Hyun-Ho Lim[2,3] & Sangho Lee [1] ✉

The anion channel SLAC1 functions as a crucial effector in the ABA signaling, leading to stomata closure. SLAC1 is activated by phosphorylation in its intracellular domains. Both a binding-activation model and an inhibition-release model for activation have been proposed based on only the closed structures of SLAC1, rendering the structure-based activation mechanism controversial. Here we report cryo-EM structures of *Arabidopsis* SLAC1 WT and its phosphomimetic mutants in open and closed states. Comparison of the open structure with the closed ones reveals the structural basis for opening of the conductance pore. Multiple phosphorylation of an intracellular domain (ICD) causes dissociation of ICD from the transmembrane domain. A conserved, positively-charged sequence motif in the intracellular loop 2 (ICL2) seems to be capable of sensing of the negatively charged phosphorylated ICD. Interactions between ICL2 and ICD drive drastic conformational changes, thereby widening the pore. From our results we propose that SLAC1 operates by a mechanism combining the binding-activation and inhibition-release models.

Land plants have adapted to dry terrestrial environments by having cuticles on their surface to suppress water loss[1]. However, the impermeability of the cuticle restricts gas exchange crucial for photosynthesis[2]. Consequently, stomata allow land plants to access aerial environment[2,3]. Stomata opening is regulated by a pair of guard cells by turgor pressure change. In the long evolutionary history, basic strategy of stomata regulation using osmotic pressure has not changed[4,5]. Stomata opening facilitates gas exchange and transpiration, but provokes water loss and pathogen entry[6–8]. Abiotic and biotic cues such as drought, light, humidity, $CO_2$, ozone and pathogens are perceived by abscisic acid (ABA), salicylic acid (SA), $Ca^{2+}$, ethylene, nitric oxide, hydrogen peroxide and hydrogen sulfide, activating pertinent signaling pathways to regulate the stomata opening[7,9–17].

Osmotic pressure change in the stomata opening is mediated by osmolyte movements through diverse ion channels and transporters[2].

Slow anion channel associated 1 (SLAC1) is a slow (S)-type anion channel that generates anion ($Cl^-$ and nitrate) efflux on the guard cell plasma membrane and acts as a key effector of stomatal closure[18,19]. Trimeric SLAC1 harbors N-terminal and C-terminal intracellular domains (ICDs) with the transmembrane domain (TMD) positioned in the middle. Signaling molecules such as ABA, $CO_2$, $Ca^{2+}$, SA, and hydrogen peroxide, participate in regulation of SLAC1 by means of phosphorylation and dephosphorylation via kinases and phosphatases[11,20–26]. Multiple serine/threonine phosphorylation in the N-terminal and C-terminal intracellular domains activates SLAC1[20,21,27,28]. Kinases responsible for such multi-site phosphorylation

[1]Department of Biological Sciences, Sungkyunkwan University, Suwon 16419, Republic of Korea. [2]Neurovascular Unit Research Group, Korea Brain Research Institute, Daegu 41068, Republic of Korea. [3]Department of Brain Sciences, Daegu Gyeongbuk Institute of Science & Technology (DGIST), Daegu 42988, Republic of Korea. [4]Electron Microscopy Research Center, Korea Basic Science Institute, Cheongju 28119, Republic of Korea. [5]Metabolic Engineering Division, National Institute of Agricultural Sciences, Rural Development Administration, Jeonju 54874, Republic of Korea. ✉e-mail: sangholee@skku.edu

include OPEN STOMATA 1 (OST1), calcium-dependent protein kinases (CPKs), calcineurin B-like/CBL interacting protein kinases (CBL-CIPKs), STRESS INDUCED FACTOR 2 (SIF2) and GUARD CELL HYDROGEN PEROXIDE-RESISTANT 1 (GHR1)[20–22,24,27,29,30]. Phosphatases responsible for the deactivation of SLAC1 by dephosphorylation include ABA-INSENSITIVE 1 (ABI1), ABI2 and protein phosphatase 2 C (PP2CA)[20–22]. Major phosphorylation sites (S59, S65, S86, S120, and T513) and several minor ones that are important for SLAC1 activation have been well defined[20,27–31]. These multiple phosphorylation of SLAC1 enables it to integrate signals from various signaling pathways. However, SLAC1 activation mechanism remains elusive due to limitation of our knowledge on how multiple phosphorylation events are coupled to conformational changes leading to the channel opening.

Despite extensive structural studies, only closed state structures of SLAC1 have been determined from a bacterial orthologue *Haemophilus influenzae* TehA (HiTehA), a monocot plant *Brachypodium distachyon* (Bd) and a dicot model plant *Arabidopsis thaliana* (At)[28,31,32]. From structural analysis of the closed SLAC1 structures, two alternative models have been suggested for how phosphorylation leads to the activation of SLAC1: the binding-activation model[28] and the inhibition-release model[31]. The binding-activation model claims that negatively charged phosphorylated serine/threonine residues of the intracellular domains (ICDs) bind to the positively charged surface of the transmembrane domain (TMD), leading to channel opening. The inhibition-release model states that ICDs work as self-inhibitory plugs that block the pore and phosphorylation of serine/threonine residues of ICDs relieves such inhibition, thereby activating SLAC1. Currently available SLAC1 structures in the closed state implicate controversy in the activation mechanism. The closed BdSLAC1 structure supports the binding-activation model because the closed BdSLAC1 does not interact with the ICD[28]. In contrast, a closed and phosphorylation-incapable mutant AtSLAC1-S59A structure reveals the ICD bound to TMD to act as a plug, supporting the inhibition-release model[31].

To reveal the activation mechanism of SLAC1 unequivocally, here we report cryo-EM structures of AtSLAC1 with different phosphomimetic states: wild-type (WT) in an open state, 6D (T62D/S65D/S107D/S124D/S146D/S152D)[29] in both open and closed states, and 8D (S59D/T62D/S65D/S86D/S107D/S124D/S146D/S152D) in a closed state. Conformational changes around the ion pore allows us to uncover a pore opening mechanism and several putative anion binding sites. A conserved, positively charged sequence motif in the intracellular loop 2 (ICL2) shows conformational changes that are essential for phosphorylation-mediated activation. The conformational changes caused by the ICL2 are coupled to partial unwinding of TM5 and helical rearrangements of other TMs. These coupled conformational changes widen the intracellular side of the pore just as in a putative open mechanism from a previous study[31]. The discovered latch motif in a closed state supports that phosphorylation sites on ICD are not randomly spread in intracellular space, but that their spatial distributions are restricted by the latch. We propose a combined activation model compatible with both the binding-activation and the inhibition-release models. These findings give us a perspective on diverse phosphorylation sites and their roles in SLAC1 regulation.

## Results

### Two distinct conformational states of SLAC1 by cryo-EM

To elucidate activation mechanism of SLAC1, we expressed and purified AtSLAC1 WT and known phosphomimetic active mutant, AtSLAC1 6D[29], to determine the AtSLAC1 structures in different conformational states (Supplementary Fig. 1a). To overcome its low expression level and solubility, we introduced a C-terminal tag comprising super folder GFP (sfGFP)-hemagglutinin (HA)-decahistidine ($H_{10}$). To evaluate the functional activities of the recombinant AtSLAC1 channels, we performed whole-cell patch clamp recordings on HEK293T cells expressing AtSLAC1 WT or 6D mutant in the presence or absence of the

C-terminal sfGFP-HA-$H_{10}$ tag (Supplementary Fig. 1b-d). HEK293T cells expressing AtSLAC1 WT evoked robust ionic currents compared to the mock-transfected cells. The ionic currents were instantaneously generated upon voltage stimuli with linear I-V relationship from −100 mV to +100 mV in symmetrical [Cl⁻] condition. AtSLAC1 6D mutant generated slightly lower current densities to the WT AtSLAC1 as recently reported SLAC1 phosphorylation by endogenous kinases in HEK293 cells[31]. The sfGFP tag did not change the current density, activation kinetics, and I-V relationship of both AtSLAC1 WT and 6D mutant channels (Supplementary Fig. 1c, d). These electrophysiological data indicate that the AtSLAC1 with sfGFP tag retained functional integrity. Split or entire fluorescent protein fused SLAC1 constructs were unproblematic to its expression, localization and function, and frequently used from previous functional studies[20–22,24,26,27,33]. Recent studies have suggested that two phosphorylation sites, S59 and S86, are crucial for SLAC1 activation[28,31]. Unexpectedly, introduction of both or either of phosphomimetic S59D or S86D mutation to the 6D mutant caused a significant decrease in current density to the level comparable with phospho-defective S59A mutant (Fig. 1a, Supplementary Fig. 1e–g). When we tested either phosphomimetic S59D or S59E mutation, both mutants showed impaired conductivity (Supplementary Fig. 1f, g). These mutations did not affect expression level and localization (Supplementary Fig. 2a). These results indicate that phosphomimetic aspartate mutations on two major phosphorylation sites, S59 and S86, poorly mimic natural phosphorylation. Nevertheless, AtSLAC1 WT and its phosphomimetic mutants with variety of activity enabled us to find out conformational changes according to their functionality (Fig.1a).

We expressed the AtSLAC1 fusion proteins in insect cells, and purified them in the presence of glyco-diosgenin (GDN) and cholesteryl hemisuccinate (CHS) (Supplementary Fig. 3–6). Since cleavage of the C-terminal tag caused severe precipitation of the purified proteins, we conducted the cryo-EM of the AtSLAC1 WT, phosphomimetic mutants 6D (T62D/S65D/S107D/S124D/S146D/S152D), 7D (6D + S59D) and 8D (6D + S59D/S86D) without the C-terminal tag removal (Fig. 1a). From all datasets, we found two types of particles that represent distinct conformations of trimer (Fig. 1b, c, Supplementary Fig. 7, 8). One type represented a state similar to the previously known closed state[31] that featured TMD-proximal ICD domains acting as an inhibitory plug, and ten intact transmembrane helices (TMs) aligned in one direction (Fig. 1b, d, Supplementary Fig. 9, 10). The other type represented a state with features as an open state: absence of ICD plug density, weak ICD density around center of trimer, partial unwound TM5 and altered direction of TM7 (Fig. 1c, e, Supplementary Fig. 9, 10). Bimodal conformational landscape from cryo-EM data is consistent with electrophysiological characterization. Since we prepared AtSLAC1 from an insect cell expression system, AtSLAC1 were likely to be phosphorylated by endogenous kinases of insect cell origin as similarly reported[31]. Highly conductive AtSLAC1 WT showed predominantly the open conformation, whereas less conductive AtSLAC1 7D and 8D exhibited mostly the closed conformation (Fig. 1a, Supplementary Fig. 1, 5, 6). Moderately conductive AtSLAC1 6D mutant adopted both the open and closed conformations (Fig. 1a, Supplementary Fig. 1, 3, 4). Interestingly, protein concentration and blotting condition appeared to affect the bimodal conformational landscape. We obtained an open conformation of AtSLAC1 6D from the grids of relatively lower sample concentrations and sparse particle distribution (Supplementary Fig. 7). In contrast, the closed conformation of AtSLAC1 6D was observed in the conditions of a higher sample concentration and dense particle distribution (Supplementary Fig. 7). We assumed that inter-particle interferences might have affected interactions between TMD and ICD crucial for activation. To reveal closed to open transition process, we focused on AtSLAC1 6D that represented two different conformations. In order to achieve better resolution from homogeneous population, we selected two datasets showing mostly open conformation to

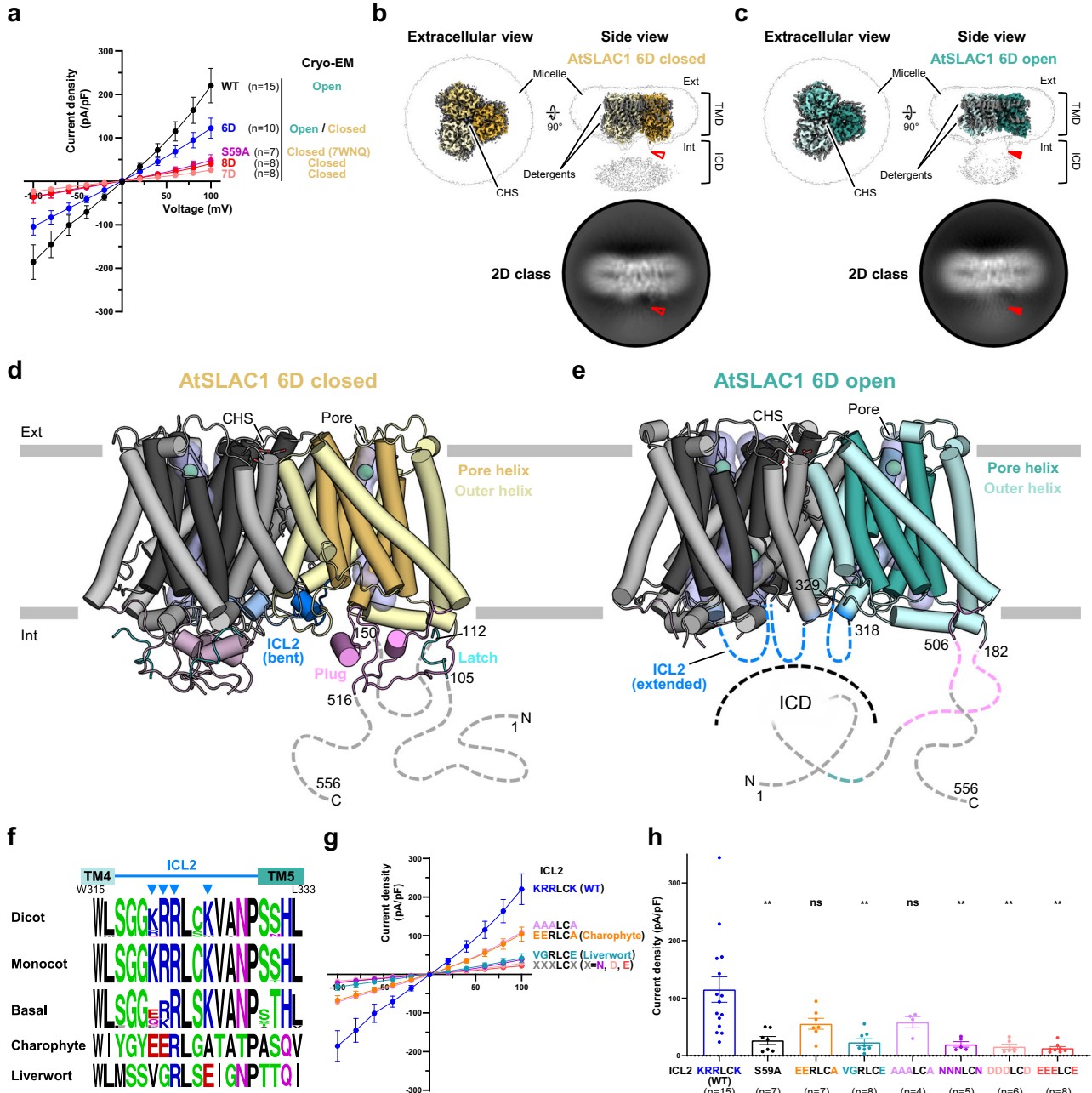

**Fig. 1 | Structures of *Arabidopsis thaliana* SLAC1 6D mutant in open and closed conformations. a** I-V relationship of AtSLAC1 wild-type (WT), S59A and phospho-mimetic mutants (6D, 7D and 8D). For each construct, open and closed states are indicated. The conformational state of S59A is based on a known structure (PDB: 7WNQ). **b**, **c** Cryo-EM maps of open (teal) and closed (yellow) conformations of AtSLAC1 6D in two views. Densities for each protomer are colored with varying brightness; those for cholesterol hemisuccinate (CHS), detergents and unmodeled regions as gray. Unsharpened maps at a lower contour level (black border lines) show densities of micelle and intracellular regions. Representative 2D class averages in side view are shown. Presence and absence of the connected densities are indicated by filled and empty red triangle, respectively. TMD transmembrane domain, ICD intracellular domain, Ext extracellular side, and Int intracellular side. **d**, **e** Open and closed structures of AtSLAC1 6D. Pore helices are colored as teal (open) and yellow (closed); outer helices and loops as cyan (open) and light yellow (closed); and ICL2s as blue. Residue numbers, N- and C-termini are indicated.

Disordered regions and ICL2 are represented as dashed lines; membrane bilayers as gray thick lines; pore volumes as blue tubes; and putative chloride ion sites as green spheres. **f** Conserved sequence motifs nearby ICL2 in dicots, monocots, basal plants, a charophyte *Klebsormidium nitens* and a liverwort *Marchantia polymorpha*. Height of each upper-case letter is proportional to conservation level. Blue inverted triangles indicate conserved lysine and arginine sites on ICL2. Sequence alignments were performed by ClustalX[54]. The plot was prepared using WebLogo[55]. **g** I-V relationship of WT ICL2 (KRRLCK) and its mutants in HEK293T cells. **h** Current densities of WT and its ICL2 mutants in (**g**) at +60 mV. In (**a**, **g**) numbers of observations are indicated in parentheses. Each data was represented as mean ± standard error. One-way ANOVA was used for comparisons against WT with others from Dunnett's T3 multiple comparisons test. *P* values are 0.0094, 0.1431, 0.0071, 0.1841, 0.0052, 0.0035 and 0.0030, respectively. Ns not statistically significant; *$P \leq$ 0.05; **$P \leq$ 0.01.

determine the open state and a dataset featuring a large proportion of the closed conformation to determine the closed state structure with C3 symmetry, both at resolution of 3.3 Å (Supplementary Fig. 3, 4, 7). We also determined the open state of AtSLAC1 WT and the closed state of AtSLAC1 8D with C3 symmetry, at resolution of 3.8 Å and 2.7 Å, respectively (Supplementary Fig. 5, 6, 8). Both closed and open states are nearly identical to those of 6D. Cα atoms root mean square deviation (Cα r.m.s.d.) between the open states was 0.58 Å, and that of closed states was 0.42 Å. The AtSLAC1 7D showed a low resolution (4.33 Å) and was not modeled due to poor grid state including ice contamination, but it also represented the closed state that was identical to others.

The key structural elements of the SLAC1 structures, as in bacterial homologue HiTehA (PDB: 3M71, 19% sequence identity of TMD)[32], monocot BdSLAC1 (PDB: 7EN0, 73% sequence identity of TMD)[28] and dicot AtSLAC1 S59A (PDB: 7WNQ)[31], are well conserved in our AtSLAC1 structures: 10 transmembrane helices (TMs) comprising the pore formed by five odd-numbered inner TMs and outer helices formed by five even-numbered TMs[28,31,32] (Fig. 1d, e, Supplementary Fig. 9). In our AtSLAC1 open and closed structures, three CHS densities were well visible and modelled in the conserved central lipid pocket on the extracellular side at the center of the trimer (Fig. 1b–e, Supplementary Fig. 3–6). We also observed ordered detergent densities in both outer and inner leaflets of TMD (Fig. 1b, c). The closed structure of AtSLAC1 6D was well aligned with that of phospho-defective AtSLAC1 S59A[31] with Cα r.m.s.d. being only 0.45 Å. N- and C-terminal plug regions were also well visible.

We found the existence of an unidentified density (hereinafter called as "latch") nearby the plug region in the cryo-EM maps of the closed structures of AtSLAC1: the maps of 6D and 8D as observed in that of S59A (PDB: 7WNQ; EMD-32633)[31] (Fig. 1d, Supplementary Fig. 11a). However, ambiguous and disconnected densities in the maps of AtSLAC1 6D and S59A[31] made it difficult to discriminate them as protein densities (Supplementary Fig. 11b). By contrast, the cryo-EM map of AtSLAC1 8D exhibited a clear density for the latch with the sequence of ${}^{105}$DF(S107D)MFRTK${}^{112}$, revealing features for unequivocal modeling: an apparent N-terminal direction due to a right-helical turn and side chain orientation; readily discernible densities of the aromatic amino acid pattern (F/Y)xx(F/Y)R; distinct density of the phosphomimetic mutation S107D compared to that of the corresponding S107 in the AtSLAC1 S59A[31] (Supplementary Fig. 11b). Additionally, S107D and S152D phosphomimetic mutations were visible from both the latch and the plug in our AtSLAC1 6D and 8D structures, but these mutations caused negligible structural differences compared to AtSLAC1 S59A[31] (Supplementary Fig. 11b, c). The latch is located nearby the plug region, but the gap between the C-terminal end of the latch (K112) and the N-terminal end of the plug (N148) is long with 35 amino acid residues (Supplementary Fig. 11c). The latch binds to both TMD and the plug with a buried surface area (BSA) of 522.7 Å$^2$ (Supplementary Fig. 11d). The binding of latch is expected to stabilize the plug by increasing interface between TMD and ICD (plug and latch) from 1176.8 Å$^2$ to 1446.9 Å$^2$ BSA. It seems that the latch acts as an inhibitory element along with the plug, as discussed later.

## Activation of SLAC1 by phosphorylation and ICL2 switch

The open state structures of AtSLAC1 revealed distinct conformational changes from closed states. AtSLAC1 6D open state was modeled with the TMD region (residues 182-318, 329-506) except the faint density of intracellular loop 2 (ICL2) (residues 319–328), and closed state was modeled with the TMD and an intracellular plug (residues 150-516) (Fig. 1d, e). Key structural elements differentiating the open and closed states of AtSLAC1 6D are revealed by the relationship between the faint densities of ICD and ICL2. Both 2D class averages and map at a low contour level of the open structure show the faint ICD density that is connected to center of trimer (Fig. 1c, Supplementary Fig. 10b). Unlike the closed structure, ICL2s in the center of trimer make contacts with ICD and are pulled away from TMD (Fig. 1d, e, Supplementary Fig. 10a, b). To exclude artificial features from enforced C3 symmetry, we also conducted refinement without symmetry in both AtSLAC1 WT and 6D open state (Supplementary Fig. 10c, d). Despite no applied symmetry, all protomers of WT and 6D open state are almost identical except ICL2 and ICD that have asymmetrical distribution as expected from weak density with C3 symmetry (Supplementary Fig. 10c, d). Nevertheless, ICL2s are extended from the center of trimer, and connected to ICD regardless of applied symmetry (Supplementary Fig. 10b, c, d). Conformations around the ICL2 are most prominent features defining the open and closed states. In the closed structure, the ICL2 lies down on the membrane plane, which hereinafter we call as "bent state" (Fig. 1d, Supplementary Fig. 10a). In the open structure, the ICD apparently pulls the ICL2 to the intracellular side, hereinafter "extended state" (Fig. 1e, Supplementary Fig. 10b). Interactions of ICL2 with ICD are likely to cause conformational changes for channel opening, which is expected from the binding-activation model.

To corroborate functional relevance of the ICL2, we examined the ICL2 region in multiple sequence alignments of orthologous SLAC1 sequences from dicot, monocot and basal plants, and two ancient SLAC1 orthologue sequences from a charyophyte *Klebsormidium nitens* (Kn) and a liverwort *Marchantia polymorpha* (Mp) based on a previous study[34] (Fig. 1f). Both KnSLAC1 and MpSLAC1 are OST1-mediated, phosphorylation-insensitive SLAC1 due to lack of OST1-specific phosphorylation sites[35]. We found a conserved amino acid sequence motif consisting of several positively charged residues (lysine and arginine) that might interact with negatively charged, phosphorylated residues of ICDs: (K/R)RRL(C/S)K in dicot and monocot plants and x(K/R)RLSK where "x" indicates any amino acid residue including aspartate, glutamate, lysine, glutamine and valine in basal plants (Fig. 1f). The net charges of ICL2 of phosphorylation-insensitive KnSLAC1 and MpSLAC1 are either negative or zero, rendering it difficult to form salt bridges with the phosphorylated residues (Fig. 1f). Lining of the positively charged residues in ICL2 seems to have evolved upon emersion of phosphorylation regulation mechanism. To verify the functional relevance of ICL2 on channel activation, we mutated these positively charged residues (KRRxxK) of AtSLAC1 to corresponding residues of phosphor-insensitive KnSLAC1 (EERxxA), MpSLAC1 (VGRxxE) and various amino acid substitutions (AAAxxA, NNNxxN, DDDxxD and EEExxE) (Fig. 1g, h). AtSLAC1 WT (${}^{320}$KRRLCK${}^{325}$) and its ICL2 mutants were expressed in HEK293T cells, and whole-cell currents from these constructs were recorded by whole-cell patch clamp recordings. ICL2 mutants showed similar or higher expression level than that of WT AtSLAC1 (Supplementary Fig. 2b). Compared to WT AtSLAC1, the ionic currents of most ICL2 mutants showed significantly reduced current density as that of phospho-defective S59A mutant, but EERLCA (KnSLAC1) and AAALCA (alanine substitution) mutants showed less reduced current density (Fig. 1g, h). Two arginine residues in ICL2, R321 and R322, are expected to be required for structural integrity of the closed structure, unlike two lysine residues K320 and K325 that are exposed to the intracellular space (Supplementary Fig. 12a). The two arginine residues interact with neighboring residues: V151, F259, Q314 and S317 in the same protomer; L372 and Q374 on adjacent protomer in the closed structures of AtSLAC1 6D, 8D and S59A (PDB: 7WNQ) (Supplementary Fig. 12a). We restored these two arginine residues from above mutants to exclude possibility that reduced current density might have come from structural collapse. The results showed that mutations on two solvent-exposed lysine residues are sufficient to abolish the activation of SLAC1 (Supplementary Fig. 12b). Also, restoration of two arginine residues from moderate conductive EERLCA and AAALCA mutants resulted in a much lower conductivity, implicating that the two arginine residues of ICL2

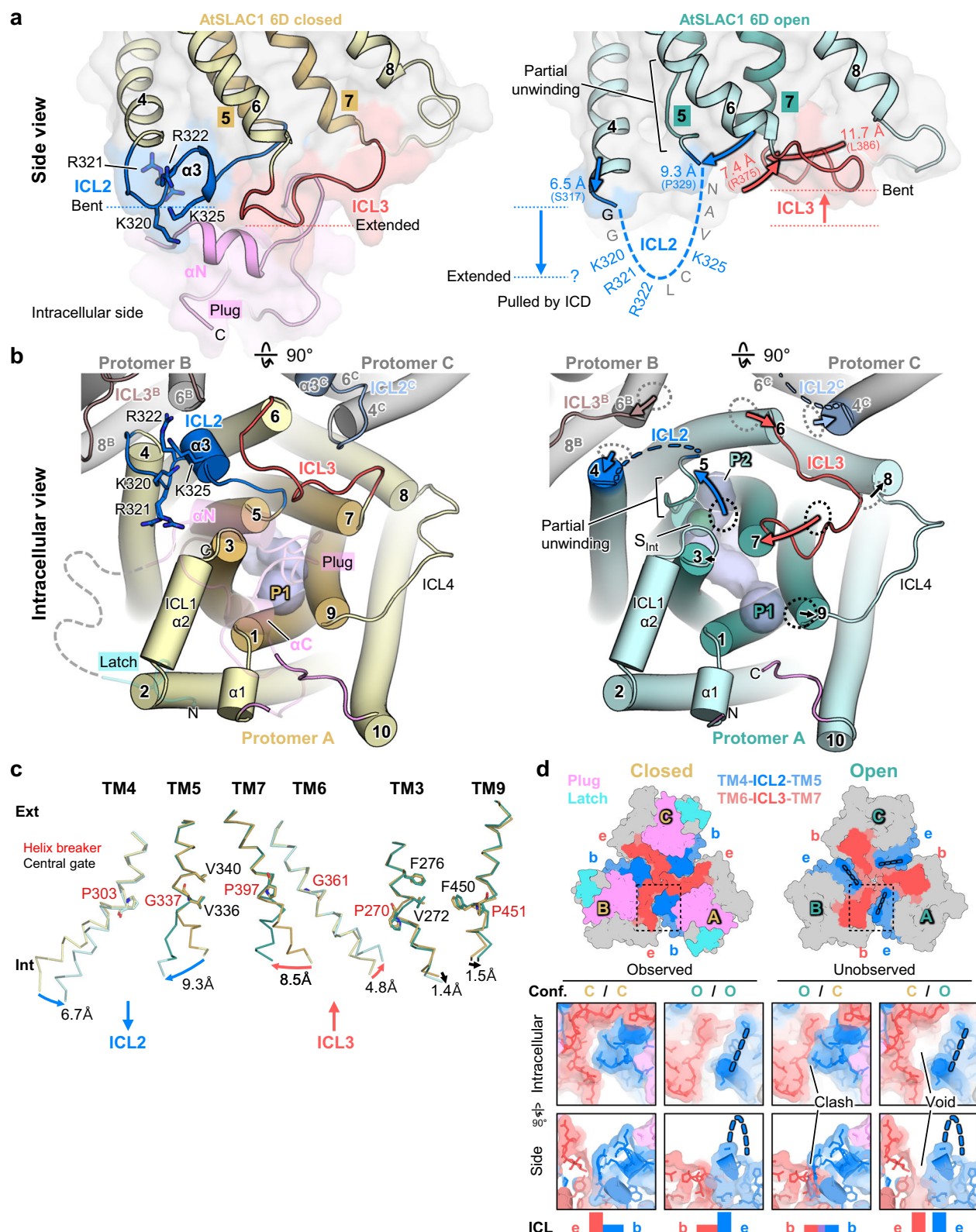

**a** AtSLAC1 6D closed / AtSLAC1 6D open

**b** Protomer B / Protomer C ... Protomer A

**c** TM4 TM5 TM7 TM6 TM3 TM9

**d** Closed / Open

are important for maintenance of closed conformation (Fig. 1g, h, Supplementary Fig. 12b). In contrast, ICL2 mutants on AtSLAC1 6D showed non-significant decrease of current density (Supplementary Fig. 12c). It seemed to reflect weak interactions between ICD and ICL2 in 6D mutant. Taken together, these results demonstrate that ICL2 is a putative activation switch that can sense the phosphorylation of SLAC1.

## Rearrangement of transmembrane helices on the intracellular side upon activation

The pulling of ICL2 results in partial unwinding of TM5, followed by distinct topological changes in the transition from the closed to open structures (Fig. 2a, b, Supplementary Fig. 9). The ICL2 transition from the bent to extended states is concomitant with pulling up its connected TM4 and TM5 toward the intracellular side (Fig. 2a). TM4 in the

**Fig. 2 | Transmembrane helices rearrangement on intracellular side caused by ICL2 movement. a** Side views of TMD of closed and open structures of 6D. The structures are depicted as cartoon and surface representations. The closed structure is colored as yellow (pore helices) and light yellow (outer helices); the open structure as teal (pore helices) and cyan (outer helices); and ICL2, ICL3 and plug as blue, red and pink, respectively. Movements of ICL2- or ICL3-connected TMs are indicated by thick arrows with distances. Disordered ICL2 is indicated by dashed lines with residues labeled. **b** Intracellular views of TMD. Protomer A is depicted as colored cartoon representations; plug and latch as transparent ones; and two other protomers (B and C) of the trimeric 6D as gray ones. Numbers indicate TM numbers with protomer identifier as superscripts; gray dashed line disordered ICD; blue dashed lines ICL2; and thick arrows TM movements of the open structure. TM4, TM6 and TM8 of the closed structure are depicted as gray dashed circles and TM5, TM7 and TM9 as black ones. Putative pathways are labeled as P1 and P2. Pore

volumes are shown as light blue tubes and putative solvent density ($S_{Int}$) as a green sphere. **c** Helix breaker-induced TM movements. TMs of the open (teal) and closed (yellow) structures are depicted as Cα traces; the helix breakers and central gate residues as stick models with red and black labels, respectively. Ext, extracellular side; int, intracellular side. **d** (*Upper*) Complementary movements of ICL2 and ICL3 at protomer interfaces viewed from the intracellular side. All models are depicted as surface representations. Coloring scheme is the same as in (**a**). Disordered ICL2s are indicated by dashed thick blue lines. Protomers are labeled as A, B and C; bent and extended states as b and e, respectively. (*Lower*) Close-ups of a protomer interface (boxed by dashed lines). Closed and open conformations ("conf.") are labeled as C (yellow) and O (teal), respectively. Relative positioning of ICL2 and ICL3 is schematically shown by rectangles as colored in (**a**); steric clash and void by purple rectangle and a gap, respectively.

open structure moves toward the intracellular side by 6.5 Å translation and 14-degree rotation compared to that in the closed structure. TM5 is partially unwound and bent away from the pore axis by 9.3 Å translation and 27-degree rotation (Fig. 2a, b). The pulling of TM4-ICL2-TM5 toward the intracellular side in the open structure causes sequential rearrangements of TMs: bending of TM7 toward TM5, bending of TM6 toward TM4 of the adjacent protomer, and slight movements of TM3 and 9 to avoid steric clashes with TM7 (Fig. 2b). Notably, these concerted helical rearrangements are only limited to the intracellular side of TMs (Fig. 2c, Supplementary Fig. 13). Previous studies suggest that highly conserved helix breakers (G198[TM1], P270[TM3], G337[TM5], P397[TM7] and P451[TM9]) in the middle of the pore helices may contribute to pore regulation by helix movements such as rotation and kink[28,31,32]. Cα r.m.s.d.s among our AtSLAC1 6D open, closed, and two known closed structures are negligible on the extracellular side of TMs based on these helix breakers (Supplementary Fig. 13a, c). By contrast, Cα r.m.s.d.s of TMs on the intracellular side between the open and closed structures are much higher than those among the closed ones, especially around TM4, 5, 6 and 7 that are connected to ICL2 and ICL3 (Fig. 2a–c, Supplementary Fig. 13b, c). Despite presence of the putative helix breaker G198, TM1 shows negligible changes during transition on both intracellular and extracellular sides because it is located far away from ICL2 and ICL3 movements (Supplementary Fig. 13b, c). In addition, we found that outer-helices TM4 and 6 that are important for cooperativity also have highly conserved helix breakers P303 and G361, respectively (Fig. 2c, Supplementary Fig. 13b). In multiple sequence alignment, these two sites are identical in all SLAC1 sequences including Kn and MpSLAC1 where TM4 and 6 sequences are less conserved. Highly conserved helix breakers with corresponding helical movements imply the involvement of the helix breakers in the transition from the closed to the open structures. Taken together, it seems that the helix breakers divide TMs to the intracellular and extracellular sides and are essential for intracellular conformational transitions such as bending and unwinding of TMs.

The ICL3 connecting TM6 and TM7 also alternates the extended and bent states, in directions opposite to the states of ICL2 (Fig. 2a–d). In the closed structure, ICL3 bulges out and contacts with the plug and αN of N-terminal ICD, but ICL3 in the open structure lies down on TMD with dissociation of the ICD plug and to fill the gap caused by TM7 and adjacent TM4 movements (Fig. 2a, b). ICL2 of one protomer and ICL3 of an adjacent protomer face each other at the protomer-protomer interface (Fig. 2b, d). The bent-ICL2 in the closed structure and the bent-ICL3 in the open structure occupy overlapping space on the same interface (Fig. 2d). By contrast, the extended-ICL2 in the open structure and the extended-ICL3 in the closed structure make a large void at the protomer-protomer interface (Fig. 2d). These observations implicate that the open and closed protomers are unlikely to exist in the same trimer at once due to steric clashes. Transition between the open and closed states of one protomer propagates to another one, enabling rapid transition between all-closed and all-open trimers.

These structural properties are consistent with our cryo-EM analysis and previous studies. In cryo-EM maps, we observed neither one-open nor two-open trimeric conformation, and only found either all closed trimeric or all open trimeric conformation. Previous studies reported cooperative opening of SLAC1 that has multiple conductance states and represents rapid transition throughout these states[31,36,37]. However, there was no structural rationale for the cooperativity of SLAC1. Our results provide a structural feature underlying the cooperative activation: the interlocked trimeric architecture of SLAC1 where a conformational state is shared among protomers by complementary movements of ICL2, ICL3, TM4 and TM6 at the protomer-protomer interface.

## Anion conductance pathway in the open and closed structures

To assess pore widening during the transition from the closed to open states, we conducted pore analysis of the open and closed structures using MOLE 2.5[38]. The radii on the extracellular side of the pore are wide enough for a chloride ion (1.8 Å radius) to pass through in both the closed and open structures along with negligible structural differences on the extracellular side (Fig. 3). By contrast, in the open structure, intracellular helical rearrangements bring significant changes to the intracellular side of the pore. Consistent with previous studies[28,31], in the closed structure, the long narrow constriction from the intracellular pore entry to the central gate is formed by hydrophobic residues such as F266, L333 and M392 (Fig. 3a). The radii throughout these constrictions are narrower than radius of a chloride ion, which make it impossible for a chloride ion to pass through (Fig. 3b). By contrast, in the open structure, the partial unwinding of TM5 from TMD permits a large cavity around the pore (Figs. 2b, 3a). Subsequent TM7 bending concomitant with slight outward movement of TM9 transforms the linear pore (P1) into the inverted Y-shaped pore with the second pore (P2) branching out from P1 (Figs. 2b, 3a).

Conformational changes of TMs lead to changes of pore lining residues on the intracellular side. For instance, L333 on TM5 moves away from the pore by partial unwinding of TM5 and M392 swings away from P1 to wide fork region, prompted by TM7 bending (Fig. 4a, b). TM9 movement also contributes to significant widening of the central gate by movement of F450, a key residue (Fig. 4c). As a result, most of the intracellular side and the central gate around F450 are significantly widened to more than 1.8 Å, which can allow the passage of a chloride ion (Fig. 3b). However, the radii of the entries of both pores, and rest of the central gate are still not wide enough for a chloride ion to pass through (Fig. 3b). The entries of both pores in the open structure are surrounded by three residues each: G194, F266 and S447 for P1, and S331, L372 and V389 for P2 (Fig. 4b, Supplementary Fig. 14). Interestingly, both entries consist of one serine residue and two hydrophobic residues (Fig. 4b). Hydrophilic serine residues on both entries seem to have a role in desolvation of chloride ion for access. In the closed structure, a gap among G194, F266 and S447 is too narrow, but movement of F266 and S447 during opening widens the P1

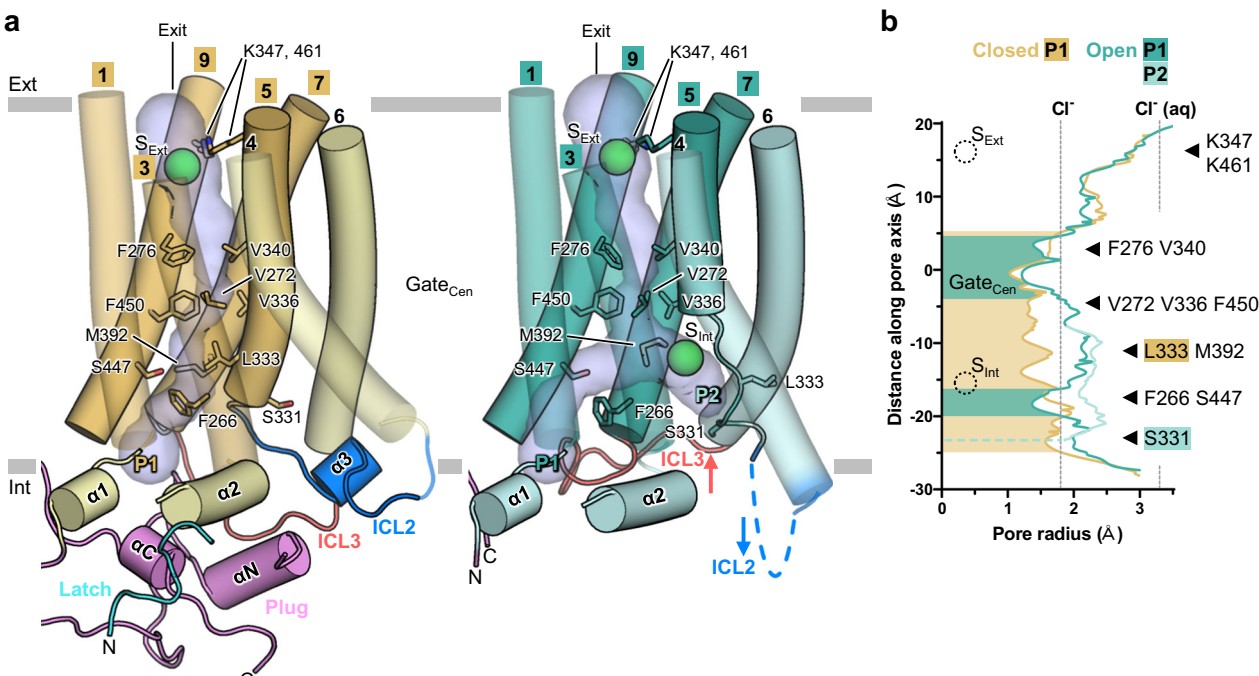

**Fig. 3 | Anion conductance pathway changes and pore opening. a** Side view of TMD in closed and open structures of AtSLAC1 6D. Only protomer A is depicted as cartoon representations. The closed structure is colored as yellow (pore helices) and light yellow (outer helices); the open structure as teal (pore helices) and cyan (outer helices). Numbers indicate TM numbers. Two putative pathways are labeled as P1 and P2, respectively. Pore volumes are shown as blue surface. Putative solvent densities ($S_{Int}$ and $S_{Ext}$) are shown as green spheres. Pore lining residues are labeled and depicted as stick models. Approximate map trace of ICL2 at a lower contour level is indicated by dashed lines. Movements of ICLs and ICD plug of the open structure in reference to those of the closed structure are indicated with black thick arrows. **b** Pore radius plot of the closed (yellow) and open (P1, teal and P2, cyan) structures. Radii for ionic (Cl⁻) and hydrated chloride ions (Cl⁻ (aq)) are indicated by gray dashed vertical lines. Constrictions smaller than ionic radius are filled with corresponding colors. Constriction of P2 by S331 is indicated by horizontal dashed line. Pore lining residues are indicated by black triangles; and those specific to each pathway by boxes with corresponding colors. Putative chloride ion sites on extracellular side of pore ($S_{Ext}$) and P2 ($S_{Int}$) are depicted as dotted circles. Central gate region is labeled as $Gate_{Cen}$. The pore radii were calculated using MOLE 2.5[38].

to 1.4 Å radius. (Fig. 4a, b, Supplementary Fig. 14a, b). P2 is formed by movement of TM5, TM6 and TM7, but three residues S331, L372 and V389 on these TMs narrow entry (Figs. 3b, 4b). Among three entry residues, side chain of S331 is exposed to solvent, so its rotameric movement enables entry to widen to 1.3 Å radius (Figs. 3b, 4b, Supplementary Fig. 14c, d). The slight loosening (from 1 Å to 1.2 Å) of the narrowest constriction site in the central gate seems to be the result of conformational changes around F276 (described later) (Figs. 3b, 4c–f). Despite narrow radii, all the pore regions along the central gate and the entries are more assessible and wider to allow the permeation of water (1.2 Å), unlike the closed state.

In the cryo-EM map, we observed putative anion densities (modeled as chloride ions due to absence of other anion in the buffer composition) in the open and closed structures: one site is located in the extracellular exit of pore, $S_{Ext}$, and the other on the intracellular side of P2, $S_{Int}$ (Supplementary Fig. 3–6). The presence of $S_{Ext}$ in both structures reflects negligible conformational differences on the extracellular side of SLAC1. $S_{Ext}$ is surrounded by two lysine residues K347 and K461, along with several hydrophobic residues (V204, I279, I343, I402 and V457). The density of $S_{Ext}$ is weak and dispersed, reflecting long distance, weak electrostatic interactions between a chloride ion and K347 and K461: 5.6 Å/4.0 Å in the 6D closed; 4.4 Å/5.1 Å in the 8D closed; 4.3 Å/4.7 Å in the 6D open; 5.8 Å/5.8 Å in the WT open (Supplementary Fig. 3–6). Two positively charge residues on $S_{Ext}$ seem to attract an anion that escapes from the central gate and to help transfer of the anion into the extracellular side. By contrast, we only observed the other putative chloride ion site $S_{Int}$ on P2 in the open structure, but not in the closed structure due to the absence of P2 (Supplementary Fig. 4, 6). P2 is more hydrophilic than the hydrophobic

for the rest of the entire pore, owing to the exposed main chain atoms from the unwound TM5 and the hydrophilic side chains of S331 and N265 (Supplementary Fig. 4, 6). The putative chloride ion density is closely located to hydrophilic atoms: carbonyl oxygen (3.7 Å) of L333, Oγ (3.8 Å) of S334 and amide nitrogen (3.7 Å) of V335 (Supplementary Fig. 4). The existence of the ion density on the intracellular side of P2 also supports the high solvent accessibility of this region. Further studies are needed to investigate the permeability of the two pores.

## Conformational changes of central gate residues by helical rearrangement

Large conformational changes on the intracellular side also affect the central gate consisting of two phenylalanine residues (F276 and F450), and three valine residues (V272, V336 and V340) (Fig. 4c–e, Supplementary Fig. 15). These central gate residues belong to three pore helices: V272 and F276 on TM3, V336 and V340 on TM5, and F450 on TM9. Despite a large conformational change from unwinding TM5, V336 and V340 show only slight changes between the closed and open structures (Fig. 4c, Supplementary Fig. 15). These slight changes may be due to the flexible helix breaker G337 by preventing the propagation of unwinding. By contrast, the rest of the central gate residues undergoes significant conformational changes affected by slight TM3 and 9 movements caused by TM7 bending (Fig. 4c–e, Supplementary Fig. 15). F276 on TM3 and F450 on TM9, the key residues in the gating mechanism, would presumably move during opening of the channel because of their conservation across species and energetically unfavorable conformation due to tight packing in the closed structure[28,32]. In the closed structure of AtSLAC1 6D, F276 on TM3 adopts an unfavorable rotameric state (−83° of χ1, 6° of χ2) in a manner similar to

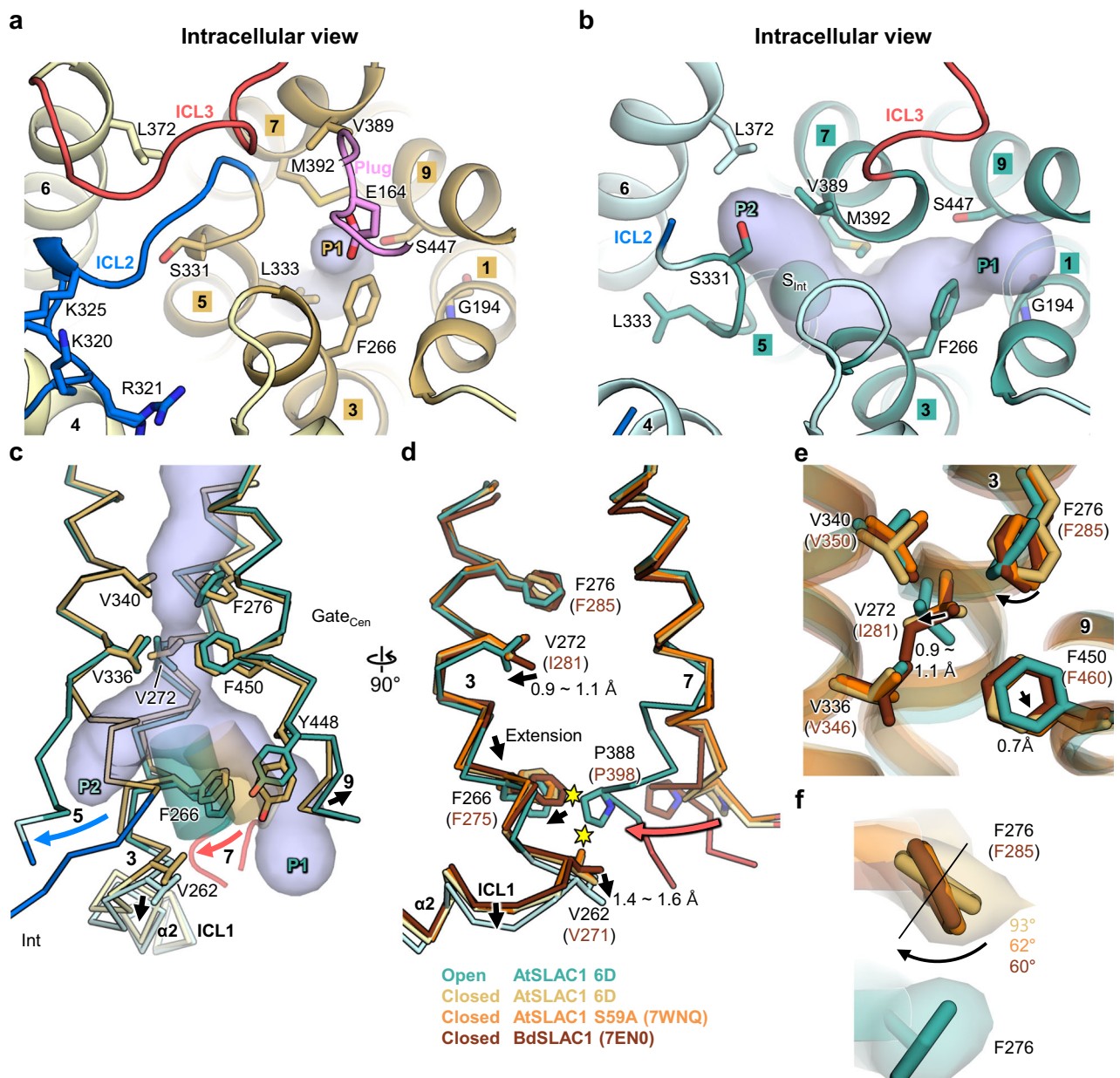

**Fig. 4 | Conformational changes on the intracellular side.** In all panels, closed structure is colored as yellow (pore helices) and light yellow (the rest) and open structure as teal (pore helices) and cyan (the rest). **a, b** Intracellular view of the closed (**a**) and open (**b**) structures of AtSLAC1 6D. Important pore residues of the closed structure and entry residues of the open structure are labeled and shown as stick models. Two putative pathways are labeled as P1 and P2, respectively. Pore volumes are shown as blue tubes. **c** Conformational changes related to central gate opening in the closed-to-open transition of AtSLAC1 6D. Movements and rotations of the open structure in reference to those of the closed structure are indicated with thick arrows. **d** Conformations of TM3 in the open and closed structures. The closed structures of AtSLAC1 S59A (PDB: 7WNQ) and BdSLAC1 (PDB: 7EN0) are colored as orange and brown, respectively. The central gate and important residues are shown as stick models. Steric clashes between the open and closed structures are indicated by yellow hexagrams. Black arrows show the direction of either helices or residues movements. **e** Comparison of the central gate residues of the open and closed SLAC1 structures. **f** Side chain rotamers of F276 of the closed and open structures. Thin black line marks position of F276 rotamer of the open structure. Rotations from the F276 rotamer of the open structure to each of the rotamer of the closed structures are shown with angles.

those in the previously reported structures, F285 of BdSLAC1 (−91° of χ1, −16° of χ2) and F276 of AtSLAC1 S59A (−86° of χ1, −22° of χ2) (Fig. 4e, Supplementary Fig. 15b, d). In the open structure, P388 on bended TM7 clashes with V262 and F266 on the TM3, and causes TM3 extension (Figs. 2c, 4c, d). Changes on TM3 are linked to the retreat of V272 (0.9 Å) from F276 and subsequent rotameric changes of F276 about an 93° (Fig. 4d, e, Supplementary Fig. 15b–e). The rotation of the phenyl moiety of F276 by 60 to 93° allows F276 to adopt an energetically favorable rotameric state (−96° of χ1, −80° of χ2) in the open

structure (Fig. 4e, f). F450 in the open structure remains in an unfavorable rotameric state (154° of χ1, 68° of χ2) which is similar to that in the closed state structures: BdSLAC1 F460 (159° of χ1, 60° of χ2), AtSLAC1 S59A F450 (153° of χ1, 71° of χ2) and AtSLAC1 6D F450 (163° of χ1, 65° of χ2). Upon bending of TM7, P388 clashes with Y448 on TM9 and steric hindrance causes bending (1.5 Å) of the intracellular side TM9 (Figs. 2c, 4c). TM9 has a kink introduced by helix breaker P451 in the closed structure (Fig. 2c). Previous studies predicted that conformational changes upon kink of TM9 may lead to changes of F450

and P451, followed by opening of the central gate[28,31]. Such a prediction is supported by the results that either alanine or glycine substitution of P451 represents impaired channel conductance[28,31]. These reported results are consistent with analysis of our open structure. Bending of TM9 causes a 4° increase of kink on helix breaker P451 (Fig. 2c). Such a kink on TM9 leads to a slight movement of F450 (0.7 Å) away from that central gate (Fig. 4e, Supplementary Fig. 15b–e).

How are these movements linked to the central gate widening? In the closed structure, constriction of the central gate is caused by two sites: a triad of V272, V336 and F450 on the intracellular side (hereinafter called as "triad constriction"), and a quad of F276, V336, V340 and F450 in the center (hereinafter called as "quad constriction") (Figs. 3b, 4c, e, Supplementary Fig. 15). Slight movements away from the pore by V272 (0.9 Å) and F450 (0.7 Å) contribute to widening of the triad constriction from radii of 1.3 Å in the closed structure to radii of 1.8–2 Å in the open structure (Figs. 3b, 4c–e, Supplementary Fig. 15). Restrained small movements of V272 and F450 compared to large movements of the intracellular side residues may have resulted from nearby helix breaker residues P270 and P451, which enables sophisticated widening of the triad constriction so that the pore size is comparable to the radius of a chloride ion. Despite retreats of V272 and F450, the widened pore of the quad constriction is filled with F276 by its rotation (Fig. 4e, Supplementary Fig. 15b, c), leading to the limited relief of quad constriction from 1 Å to 1.2 Å (Fig. 3b). However, release of the steric strain of F276 may pave the way for a complete opening. It is known that small oscillation of bulky aromatic ring can cause sufficient changes of pore dynamics due to its oblate shape[39]. In the closed structure, F276 represents an unfavorable conformation due to the confined space. Compared to the closed structure, the expanded space adjacent to F276 could permit small oscillation of χ1 and χ2 angles of F276, thereby widening the central gate transiently.

### Restriction of phosphorylation sites by plug and latch

Recent studies suggest that AtSLAC1 harbors two major phosphorylation sites S59 and S86, and two patches (patch 1 and 2) that are crowded with minor phosphorylation sites[28,31]. Our structures uncover that distribution of phosphorylation sites in the intracellular space is restricted by spatial constraints from the ordered plug and latch (Fig. 5a). To confirm whether the restricted spatial distribution of phosphorylation sites, plug and latch of AtSLAC1 are general features of SLAC1, we conducted multiple sequence alignments of orthologous SLAC1 sequences from dicot, monocot and basal plants, and two ancient SLAC1 orthologues (Supplementary Fig. 16). All plant SLAC1s except the two ancient SLAC1 orthologues have the well conserved N- and C-terminal plug (Supplementary Fig. 16). The latch is well conserved in dicot and monocot SLAC1s, and also found in some basal plant SLAC1s (Fig. 5b, Supplementary Fig. 16). In the N-terminal half of the latch sequence motif, aromatic residues F106 and F109 in the latch interact with both the plug and TMD at once, with larger BSAs (121.31 Å² and 179.19 Å²) than other residues (Fig. 5b, c). The motif featuring the two aromatic residues (F/Y)xxF is highly conserved, reflecting its importance for binding (Fig. 5b). By contrast, solvent exposed residues D105, S107 and M108 are less conserved in the latch of dicot and basal plant SLAC1s (Fig. 5b, c). Interestingly, the C-terminal half of the latch ([110]RTK[112]) is conserved despite small BSAs (Fig. 5b). We found a conserved pattern that the OST1-mediated phosphorylation motif ([110]RxxS[113])[40] spans the C-terminal half of the latch and the first serine (S113) of the patch 1 (Fig. 5a, b, Supplementary Fig. 16).

Multiple sequence alignment uncovered that the latch and the patch 1 motifs are juxtaposed and the OST1-mediated phosphorylation motif overlaps with both the latch and patch 1 motifs (Fig. 5b). Structural analysis revealed that AtSLAC1 8D represented a low conductivity closed state with a clear latch density whereas AtSLAC1 6D seemed to be in a conformational equilibrium between the open and closed states with a weak latch density (Supplementary Fig. 11b). Therefore, we

hypothesized that either latch disruption or patch 1 phosphorylation is effective in the activation of AtSLAC1 8D, but not in 6D. To disrupt latch binding, we substituted two conserved phenylalanine residues for alanine (F106A/F109A) or eight residues of the entire latch motif for glycine (8G). To test phosphorylation effects on the patch 1, we introduced the phosphomimetic mutations (S113D/T114D/S116D/S120D), called 4D. F106A/F109A and 4D, but not 8G, in the background of AtSLAC1 8D increased current densities compared to AtSLAC1 8D itself (Fig. 5d). By contrast, none of these three mutants in the background of AtSLAC1 6D showed noticeable increase in the current densities. Since all the latch or patch 1 mutants represented higher expression levels than AtSLAC1 WT, 6D and 8D (Supplementary Fig. 2a, c), we cannot exclude the possibility that the increased current densities by the F106A/F109A and 4D mutants in the background of AtSLAC1 8D are caused by factors other than the inhibition of latch binding. Taken together, multiple sequence alignment, structural and electrophysiological analyses suggest that the plug and the latch are common inhibitory components of SLAC1 in plants. Further studies are necessary to corroborate functional interplay among the latch, patch 1 and phosphorylation.

## Discussion

Phosphorylation in the N- and C-terminal ICDs of SLAC1 is essential for activation[20–22,27]. Two models have been proposed for how phosphorylation leads to the activation of SLAC1: the binding-activation model[28] and the inhibition-release model[28,31]. In the binding-activation model, negatively charged phospho-serine/threonine residues of ICDs bind to the positively charged surface of TMD, causing conformational changes for channel opening. The closed BdSLAC1 structure supports the binding-activation model based on the observation that the closed BdSLAC1 does not interact with the ICD[28]. In the inhibition-release model, intramolecular binding of ICDs to the rest of SLAC1 locks SLAC1 in a self-inhibitory state and subsequently phosphorylation of ICDs relieves SLAC1 of such inhibition, leading to the activation[31].

Our structural and electrophysiological results implicate that AtSLAC1 constructs were likely to be phosphorylated by unknown endogenous kinases of insect and human cells as previously reported[31]. Such phosphorylation by kinases from heterologous origins may have led to heterogeneity in phosphorylation states, which in turn would complicate the interpretation of phosphorylation effects by phosphomimetic mutants. We also observed that some aspartate or glutamate mutations were phospho-defective (Supplementary Fig. 1f). These observations could impose uncertainty in the interpretation of the effect of each prospective phosphorylation site. In *Xenopus* oocyte expression system, SLAC1 alone is electrically silent and activated only by co-expression of cognate kinases[20,21,41]. Electrophysiological characterization of SLAC1 mutants with cognate kinases in oocytes might be better suited to disentangle subtle and complicated phosphorylation effects. Further studies would be needed under precise control of phosphorylation state to unequivocally confirm the effects of phosphorylation on SLAC1. Nonetheless, our cryo-EM structures combined with electrophysiological data enabled us to unveil the important structural features for SLAC1 activation.

From our cryo-EM structures, we found structural features supporting both the binding-activation and the inhibition-release models. In the closed state structures, the plug consisting of the N- (residues 148-182) and C- (506-517) terminal regions of ICD and the latch (residues 105-112) make contacts with TMD in similar manners despite different phosphorylation states. Phosphorylation sites are located in the disordered region of ICD except S107, S152 and T513 on the plug[28,29,31] (Supplementary Fig. 11c, 12). In the open state, the plug and latch are released upon phosphorylation on multiple serine/threonine residues in the disordered region of ICD. However, in data processing of AtSLAC1 6D, we observed populations with closed-like structures representing a plug-free closed TMD, reminiscent of the closed

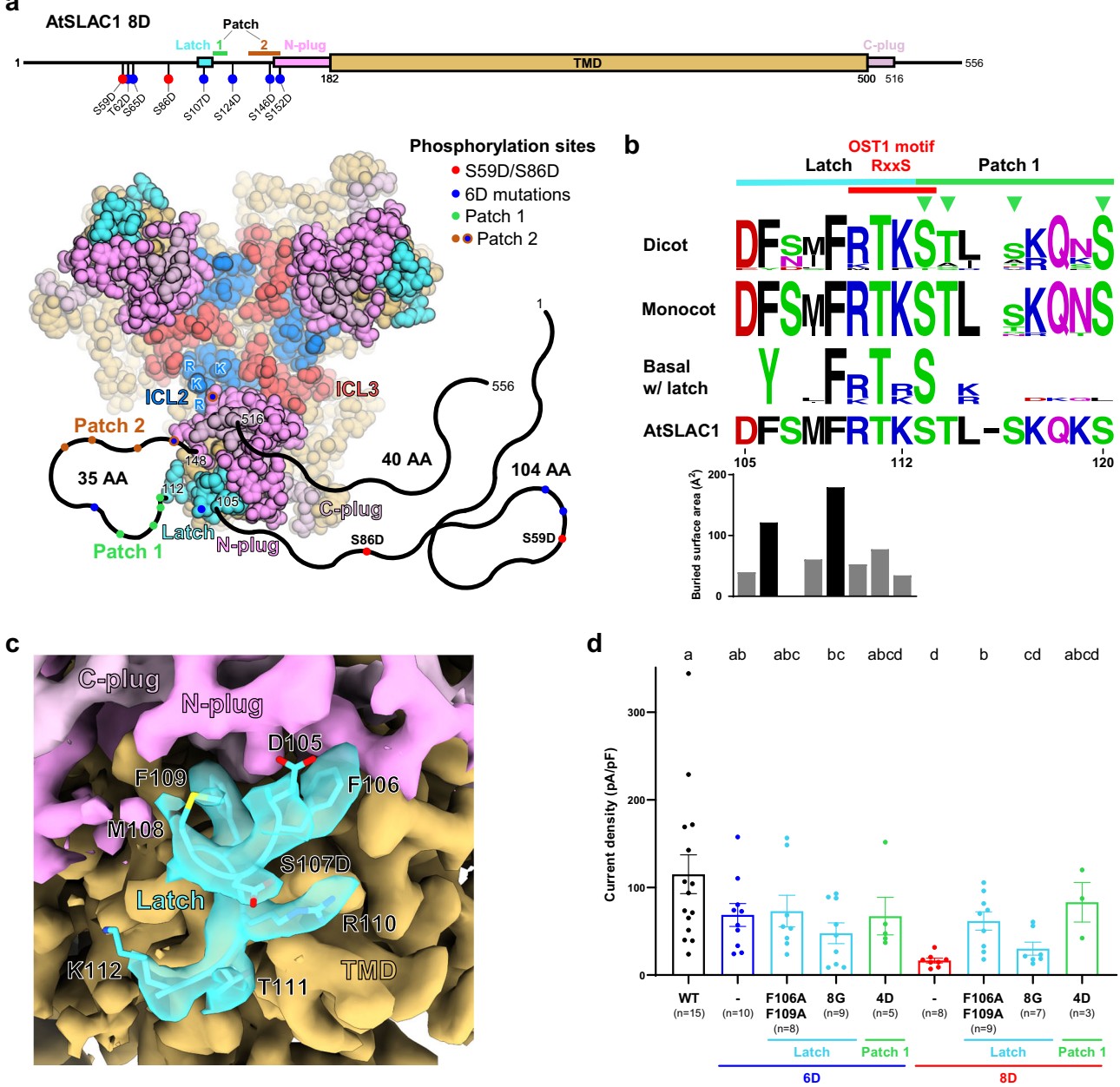

**Fig. 5 | Restriction of phosphorylation sites by plug and latch. a** Domain architecture and structural elements associated with phosphorylation sites of AtSLAC1 8D. The domain architecture is shown on top and the tertiary structure viewed from the intracellular side on bottom. Phosphorylation sites including patch 1 (green), patch 2 (brown), six phosphomimetic mutations comprising 6D (blue) and two additional mutations introduced in 8D (red) are colored as indicated. Structural elements such as latch (cyan) and N-terminal plug (pink), C-terminal plug (pale pink), ICL2 (light blue) and ICL3 (light red) are shown as spheres. TMD excluding ICL2 and ICL3 is depicted as tan box and spheres. Disordered regions which were not modeled are described as thick black lines. The number of amino acid residues in each disordered region is noted. **b** (*Upper*) Sequence conservation of latch, patch 1 and OST1 phosphorylation motif in dicots, monocots and some basal plants with latch. Height of each upper-case letter is proportional to conservation level. Green inverted triangles indicate conserved

phosphorylation sites on the patch 1 of dicot and monocot. Sequence alignments were performed by ClustalX[54]. The plot was prepared using WebLogo[55]. (*Lower*) Buried surface area per residue of AtSLAC1 8D latch is calculated by PISA[52]. The aligned amino acid sequence of AtSLAC1 is shown at the bottom with a gap. **c** Cryo-EM density around the latch of AtSLAC1 8D. The latch is shown as transparent density with stick models. TMD, N-plug, C-plug and latch are colored in the same as in (**a**). Latch residues are labeled. **d** Current densities of AtSLAC1 WT and its mutants in HEK293T cells at +60 mV. Data for mutants in the latch colored as cyan, in the patch 1 as green, in the 6D as blue and in the 8D as red. Each data was represented as mean ± standard error of the mean. Numbers of observations are indicated in parentheses. One-way ANOVA was used for comparisons against each other followed by Welch's correction. Groups sharing the same lowercase letters indicate no statistically significant difference (*P* value > 0.05) among them.

BdSLAC1 structure (Supplementary Fig. 7). AlphaFold2 prediction of AtSLAC1 (https://alphafold.ebi.ac.uk/entry/Q9LD83) also apparently features a closed TMD with a partially unfolded plug[42,43]. The ICL2 was predicted to maintain a bent state despite a negligible interaction toward the plug. These results suggest that phosphorylation events,

presumably inducing destabilization and subsequent dissociation of the plug and latch regions, are necessary for SLAC1 to escape from inhibition, reflecting the inhibition-release model, but not sufficient to convert it into the open state. In our open structures, phosphorylation events are connected to the channel opening via interactions between

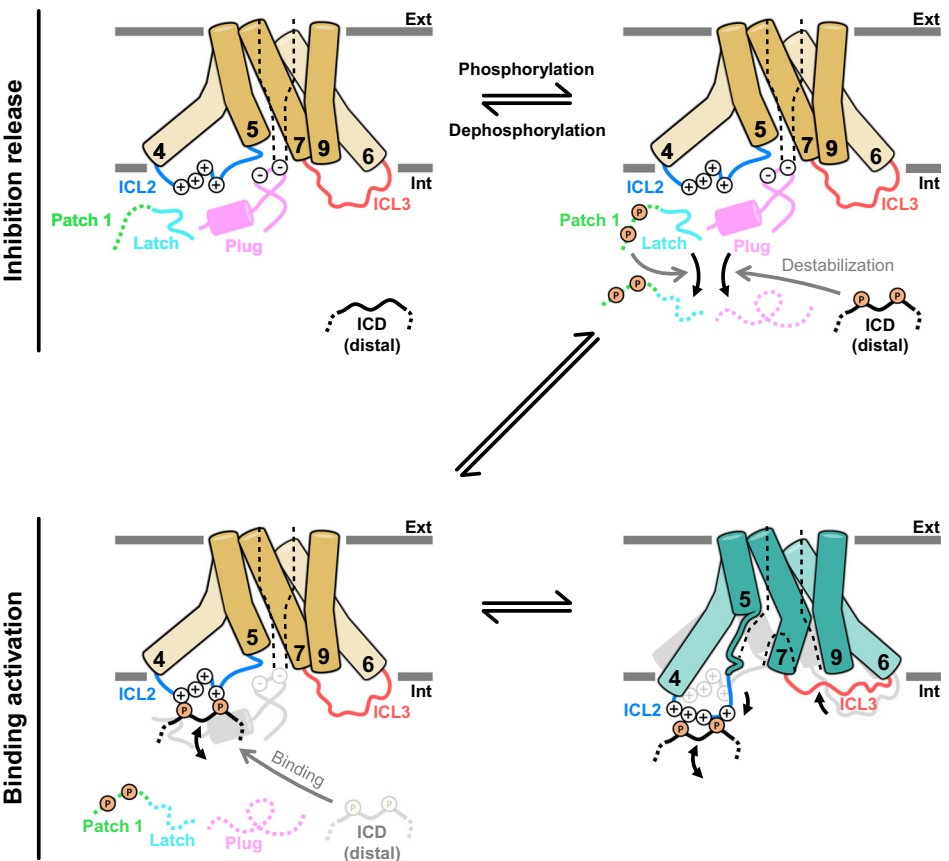

**Fig. 6 | A combined activation mechanism of AtSLAC1.** The combined activation model consists of the initial dissociation phase (inhibition-release) and the subsequent activation phase (binding-activation). In the dissociation phase, phosphorylation of the TMD-distal ICD causes destabilization of the plug and latch. In the activation phase, negatively charged phospho-serine/threonine residues on the TMD-distal ICD are perceived by the positively charged ICL2, leading to the intracellular helical rearrangement. TMD in the closed state is colored in yellow and that in the open state in teal; and TMD in the closed state when overlapped with that in the open state in pale gray. ICL2 in blue and ICL3 in red; the plug in pink, the latch in cyan and the patch 1 in green. Numbers indicate TM numbers. The positively charged residues are denoted as pluses in open circles and the phosphorylated residues as P in yellow filled circles. Thick curved black arrows indicate plausible conformational changes. Unstructured regions are depicted as dotted thick lines in corresponding colors. The pore is shown as black dotted lines. Membrane is described as two gray lines. Ext extracellular side, and int intracellular side. See the text for details.

the phosphorylated ICD and ICL2, a feature of the binding-activation model (Fig. 1). The negative charges imposed by the phosphate groups attract the cluster of positively charged residues in ICL2, with concomitant unwinding of TM5, bending of TM7 and following intracellular rearrangements rendering the reshaping into inverted Y-shaped pore and widening of pore (Figs. 2, 3).

We propose a speculative activation mechanism combining elements from the binding-activation and the inhibition-release models. The combined activation model consists of the initial dissociation phase (inhibition-release) and the subsequent activation phase (binding-activation) (Fig. 6). In the dissociation phase, phosphorylation of the TMD-distal ICD causes destabilization of the plug and latch. The distal ICD is connected to dissociation of plug from TMD, but TMD remains in the closed state. In the subsequent activation phase, negatively charged phospho-serine/threonine residues on the TMD-distal ICD are perceived by the positively charged ICL2. Such interactions between the phosphorylated residues and ICL2 lead to the intracellular helical rearrangement and the opening of channel.

The ICL2-mediated sensing and activation mechanism also suggests a structural rationale for the trimeric architecture of SLAC1 that is common from its bacterial homologue to higher plant ones[28,31,32]. SLAC1 and its homologues possess a pore on each protomer, but reported high cooperative opening indicates interprotomer relationship upon activation as previously suggested[31,36,37]. In trimer, each three ICL2s and ICL3s are alternately arranged and face each other

(Fig. 2d). Owing to limited space on protomer-protomer interface, conformational transition of ICL2 (extended or bent) is connected to complementary ICL3 movements (bent or extended) of adjacent protomer, and vice versa (Fig. 2d). ICL2 and ICL3 in same protomer also represent complementary relationship due to helical rearrangement (Fig. 2a, b). Interlocked trimeric architecture and complementary relationship between adjoining ICL2 and ICL3 leads to a domino effect that allows rapid synchronization of conformational state among protomer that enables cooperative activation. In addition, clustered ICL2s in the center of trimer are likely to contribute to cooperativity (Fig. 2d, Supplementary Fig. 10). Distances between N-terminal of TMD to the farthest ICL2 are shorter than 60 Å, which allows all ICDs in trimer to access the clustered ICL2s. Such a synchronized access of the all ICDs to the clustered ICL2s may enable trimer to recognize and integrate different phosphorylation states of each protomer at once.

Our results corroborate emerging roles of the pore-blocking phenylalanine residues. The bacterial homologue HiTehA structure has the gate residue F262 (corresponding to AtSLAC1 F450) that blocks wide pore[32]. F262A showed higher conductivity, but structure was almost identical to HiTehA WT. Several studies confirmed that equivalent F450A or corresponding mutation represented higher conductivity similar to that of bacterial homologue HiTehA[28,32,35]. Therefore, the F450 of plant SLAC1 has simply been regarded as a pore blocking residue, like its ancestor. However, recent plant SLAC1 structures represented that intracellular side of pore is

narrowed by pore residues including bulky phenylalanine residues F266, F276 and F450[28,31]. Either F276A or F450A mutant showed increased conductance without phosphorylation, presumably due to pore dilation from closed state[28]. By contrast, when phosphorylated, F266A, F276A or F450A mutant features decreased conductance despite of the pore dilation[28,31]. Thus, further conformational changes upon opening that differ from closed state were expected[28,31]. These are consistent with conformational changes that we uncovered during closed to open transition. F266 has a crucial role for TM3 extension by sterically blocking TM7, followed by the central gate changes (Fig. 4c−e). F266A may lead to insufficient steric hindrance to induce helical extension of TM3. The less steric hindrance by F266A is apparently linked to the impairment of the central gate by F276 rotation (Fig. 4d, e). F276A and F450A mutation is also expected to cause structural impairment. F276 and F450 are surrounded by hydrophobic residues. SLAC1 has deformable intracellular side that allow helical rearrangement. Cavity derived from F276A and F450A mutations may induce unexpected structural changes differing from that of open state.

Our structural analyses can provide a rationale for the discriminative behavior of SLAC1 in monocots by isoleucine substitution. The central gate residue V272, equivalent to I281 of BdSLAC1, is known to be a determinant of nitrate-dependent gating between monocot (isoleucine) and dicot (valine)[34]. Isoleucine substitution shows nitrate-dependent gating, but valine substitution shows nitrate-independent gating. In our structures, V272 induces change of the rotameric state of F276, a key residue of opening despite a slight movement by itself. In the open state of a monocot SLAC1, Cδ1 atom of I281 differing from valine may narrow the triad constriction in the central gate (Fig. 4e). It is plausible that isoleucine and valine substitution on this site among monocot and dicot plants may lead to change in ion selectivity and gating behavior. Triad constriction may act as a selectivity filter, along with similar radii of triad constriction in the open state with chloride ionic radius.

Our results raise questions about role of various phosphorylation sites. Multiple phosphorylation events on the two major sites and patch 1 are responsible for activation of SLAC1[28]. Previously, these phosphorylation sites were regarded as activators with a similar role in single activation mechanism. Our proposed combined activation model challenges whether each phosphorylation site contributes to either or both of the inhibitory component dissociation or the ICL2-mediated activation. The discovery of the ordered plug and latch from the disordered ICD may give a hint about the role of phosphorylation sites.

## Methods
### Constructs
The coding region of the full-length *Arabidopsis thaliana* SLAC1 (locus AT1G12480; UniProt Q9LD83) was cloned into a modified version of pACEBac2 (Geneva Biotech) containing a C-terminal tobacco etch virus (TEV) protease cleavage site, a HA tag (YPYDVPDYA) and a deca-histidine tag ($H_{10}$). To improve solubilization efficiency and protein stability, superfolder GFP (sfGFP)[44] was introduced in the downstream of the TEV protease cleavage site. We achieved a higher protein yield through the replacement of the hexahistidine tag with a decahistidine tag. A phosphomimetic mutant 6D was prepared by a megaprimer PCR method. Other mutations used in this study, were introduced by site-directed mutagenesis. The final constructs, pACEBac2-(AtSLAC1-WT or mutants)-sfGFP-HA-$H_{10}$ plasmids, were used to produce the corresponding recombinant baculoviral genomes.

### Recombinant baculovirus production
To produce recombinant baculoviral genomes, expression vectors were transformed into chemically competent *E. coli* DH10EmBacY (Geneva Biotech) strain. After 2 days, recombination clones were selected from blue-white screening followed by colony PCR. Bacmids

were gently extracted by midi-prep due to their fragility. Subsequently, freshly prepared Sf9 cells (11496015, Gibco) ($2.5 \times 10^6$ cells mL$^{-1}$, >90% viability) in 30 mL Sf-900 II SFM (Gibco) were transfected with 12.5 μg of recombinant bacmid using ExpiFectamine Sf transfection reagent (Gibco). Transfected cells were grown at 27 °C for 4 days and centrifuged at 1000 g for 10 min. The supernatant was filtered through 0.2 μm cellulose acetate filter, followed by the addition of 0.1% (w/v) bovine serum albumin to prevent proteolysis of recombinant P0 baculovirus and stored at 4 °C until use. Viral titer was measured by qPCR. To achieve high protein expression level, we produced P1 virus by infection of $1 \times 10^6$ Sf9 cells mL$^{-1}$ with P0 virus at a multiplicity of infection (MOI) of 0.1.

### Protein expression and purification
Recombinant baculoviruses encoding sfGFP-fused AtSLAC1-sfGFP-HA-$H_{10}$ WT and phosphomimetic mutants were used to produce recombinant proteins. Sf9 cells at $2 \times 10^6$ cells mL$^{-1}$ in 1.4-2.1 L Sf-900 II SFM were infected with recombinant P1 virus at a MOI of 5.0 and grown at 27 °C for 66 h. After expression, fluorescence of YFP and GFP was used to confirm infection level. Expressed cells were harvested at 3300 g for 10 min and kept at −80 °C until use. For lysis of cells, frozen cell pellets were resuspended in a hypotonic buffer [20 mM HEPES pH 7.0, 1 mM Tris (2-carboxyethyl) phosphine (TCEP), SigmaFAST EDTA-free protease inhibitor cocktail (Sigma)], incubated for 30 min on ice and centrifuged at 22,000 g, 4 °C for 1 h. Pellets were resuspended in a hypertonic buffer [50 mM HEPES pH 7.0, 1 M NaCl, 1 mM TCEP, 5% (v/v) glycerol, SigmaFAST EDTA-free protease inhibitor cocktail (Sigma)], and incubated for 1 h on ice and centrifuged at 22,000 g, 4 °C for 1 h. To solubilize the proteins, a solubilization buffer [50 mM HEPES pH 7.0, 150 mM NaCl, 20 mM imidazole pH 7.0, 1 mM TCEP, 5% (v/v) glycerol, SigmaFAST EDTA-free protease inhibitor cocktail (Sigma), 2% (w/v) *n*-dodecyl β-D-maltoside (DDM), 0.2% (w/v) cholesteryl hemisuccinate (CHS), 2 mM MgCl$_2$ and 12.5 U mL$^{-1}$ benzonase] was added to pellets, followed by gentle bounce homogenization. The lysate was solubilized by stirring at 4 °C for 3 h and centrifuged at 22,000 g, 4 °C for 1 h to remove insoluble materials. For capturing the solubilized his-tagged proteins, pre-equilibrated TALON Superflow (GE Healthcare) resin was poured into the cleared lysate and the resulting solution containing the TALON resin and the solubilized protein was stirred at 4 °C for 3 h. The lysate was applied into a gravity-flow column to discard unbound lysate, and washed with 5 column volume (CV) of wash I buffer [50 mM HEPES pH 7.0, 150 mM NaCl, 50 mM imidazole pH 7.0, 1 mM TCEP, 5% (v/v) glycerol, 0.1% (w/v) glyco-diosgenin (GDN), 0.02% (w/v) CHS], followed by 15 CV of wash II buffer [50 mM HEPES pH 7.0, 150 mM NaCl, 50 mM imidazole pH 7.0, 1 mM TCEP, 5% (v/v) glycerol, 0.01% (w/v) GDN, 0.002 % (w/v) CHS]. Elution buffer [50 mM HEPES pH 7.0, 150 mM NaCl, 250 mM imidazole pH 7.0, 1 mM TCEP, 5% (v/v) glycerol, 0.01% (w/v) GDN, 0.002% (w/v) CHS] was applied into the column, followed by addition of 0.1 mM EDTA into the eluate. Eluted proteins were enriched to a final volume of 600 μL with an Amicon Ultra-15 100 kDa centrifugal filter unit (Merck Millipore). To remove aggregates, the enriched proteins were centrifuged at 16,000 g, 4 °C for 1 h. After centrifugation, soluble supernatant was loaded on a Superdex 200 increase 10/300 GL (GE Healthcare) column with final buffer [20 mM HEPES pH 7.0, 150 mM NaCl, 1 mM TCEP, 0.01% (w/v) GDN, 0.002% (w/v) CHS]. For cryo-EM, fractions containing homogeneous protein from the Superdex 200 column were enriched to a final concentration of 2−4 mg mL$^{-1}$ using an Amicon Ultra-0.5 100 kDa centrifugal filter unit (Merck Millipore).

### Cryo-electron microscopy sample preparation and data acquisition
The purified AtSLAC1-6D-sfGFP-HA-$H_{10}$ proteins were diluted in the final buffer to a final concentration of 1.9, 2.4 and 3.3 mg mL$^{-1}$. No particle was observed in the hole at 1.6 mg mL$^{-1}$. Prior to sample

application, Quantifoil R 1.2/1.3 Au 200 or 300 (Electron Microscopy Sciences) grids were negatively glow-discharged for 60 s in a PELCO easiGlow glow-discharge cleaning system (Ted Pella, Inc). 3 μL of sample was applied, and grids were blotted (2.5 s for the 1.9 and 2.4 mg mL$^{-1}$ and 3.0 s for the 3.3 mg mL$^{-1}$, blotting force 5, 100% humidity, 4 °C), followed by plunge freezing in liquid ethane using a Vitrobot Mark IV (Thermo Fisher Scientific). The 1.9 and 2.4 mg mL$^{-1}$ datasets were collected on a Talos Arctica G2 (Thermo Fisher Scientific) operated at an acceleration voltage at 200 kV with a K3 detector (Gatan) and a BioQuantum energy filter (Gatan) at the Korea Basic Science Institute (KBSI). The 3.3 mg mL$^{-1}$ dataset was collected on a Titan Krios G4 (Thermo Fisher Scientific) operated at an acceleration voltage at 300 kV with a K3 detector (Gatan) and a BioQuantum energy filter (Gatan) at the Institute of Membrane Proteins (IMP). Movie data were collected using automated process of EPU software in electron counting mode. Total 3562 movies (2.4 mg mL$^{-1}$), 4792 movies (1.9 mg mL$^{-1}$) and 12,149 movies (3.3 mg mL$^{-1}$) were recorded and fractionated into 50 frames with total dose of 50 e-/Å$^2$, a pixel size of 0.84 Å per pixel (1.9 and 2.4 mg mL$^{-1}$) and a pixel size of 0.85 Å per pixel (3.3 mg mL$^{-1}$), and defocus range of −1.0 to −1.9 μm (2.4 mg mL$^{-1}$), −0.8 to −1.7 μm (1.9 mg mL$^{-1}$) and −0.7 to −1.9 μm (3.3 mg mL$^{-1}$), respectively.

The purified AtSLAC1-WT-sfGFP-HA-H$_{10}$ were diluted in the final buffer to a final concentration of 1 mg mL$^{-1}$. Prior to sample application, Quantifoil R 1.2/1.3 Au 300 (Electron Microscopy Sciences) grids were negatively glow-discharged for 60 s in a PELCO easiGlow glow-discharge cleaning system (Ted Pella, Inc). 3 μL of sample was applied, and grids were blotted with similar condition (2.5 s, blotting force 5, 100% humidity, 4 °C), followed by plunge freezing in liquid ethane using a Vitrobot Mark IV (Thermo Fisher Scientific). The datasets were collected on a Talos Arctica G2 (Thermo Fisher Scientific) operated at an acceleration voltage at 200 kV with a K3 detector (Gatan) and a BioQuantum energy filter (Gatan) at the KBSI. Total 2662 movies were recorded and fractionated into 50 frames with total dose of 50 e-/Å$^2$, a pixel size of 0.84 Å per pixel and defocus range of −0.7 to −1.9 μm.

The purified AtSLAC1-7D-sfGFP-HA-H$_{10}$ were diluted in final buffer to a final concentration of 2.1 mg mL$^{-1}$. Prior to sample application, Au-Flat 1.2/1.3 Au 300 (Protochips) grids were negatively glow-discharged for 60 s in a PELCO easiGlow glow-discharge cleaning system (Ted Pella, Inc). 3 μL of sample was applied, and grids were blotted with similar condition (2.5 s, blotting force 3, 100% humidity, 4 °C), followed by plunge freezing in liquid ethane using a Vitrobot Mark IV (Thermo Fisher Scientific). The datasets were collected on a Talos Arctica G2 (Thermo Fisher Scientific) operated at an acceleration voltage at 200 kV with a K3 detector (Gatan) and a BioQuantum energy filter (Gatan) at KBSI. Total 11,523 movies were recorded and fractionated into 50 frames with total dose of 50 e-/Å$^2$, a pixel size of 0.84 Å per pixel and defocus range of −0.8 to −1.8 μm.

The purified AtSLAC1-8D-sfGFP-HA-H$_{10}$ were diluted in the final buffer to a final concentration of 2.3 mg mL$^{-1}$. Prior to sample application, Au-Flat 1.2/1.3 Au 300 (Protochips) grids were negatively glow-discharged for 60 s in a PELCO easiGlow glow-discharge cleaning system (Ted Pella, Inc). 3 μL of sample was applied, and grids were blotted with similar condition (2.5 s, blotting force 3, 100% humidity, 4 °C), followed by plunge freezing in liquid ethane using a Vitrobot Mark IV (Thermo Fisher Scientific). The datasets were collected on a Titan Krios G4 (Thermo Fisher Scientific) operated at an acceleration voltage at 300 kV with a K3 detector (Gatan) and a BioQuantum energy filter (Gatan) at IMP. Total 11,112 movies were recorded and fractionated into 50 frames with total dose of 50 e-/Å$^2$, a pixel size of 0.858 Å per pixel and defocus range of −0.7 to −1.9 μm.

### Image processing, model building and refinement

All image processing was performed using CryoSPARC[45]. Each dataset was evaluated by data processing with a randomly sampled subset from the dataset. From initial screening and processing, we observed that particles predominantly represent open conformation in two datasets of low concentration samples (1.9 and 2.4 mg mL$^{-1}$). Therefore, we combined these datasets and used to determine the open structure. By contrast, particles representing closed conformation were observed in the datasets of relatively high concentration samples. The dataset of 3.3 mg mL$^{-1}$ sample was used to determine the closed structure. All movies were motion-corrected by patch motion correction and processed by patch CTF estimation. Next, randomly selected small subsets were used to train particle picking models. In detail, initial blob picking and following 2D classification were performed for template picking. The particles from template-based picking were 2D classified, and then selected particles were used to train a Topaz picking model. After particle extraction by the trained Topaz model, these particles were sorted by 2D classification, ab initio reconstruction, heterogeneous refinement and used to improve the Topaz model. Particles were extracted from all micrographs using the improved Topaz model. These particles were classified by 2D classification and heterogeneous refinement with an ab initio model, decoys and junks. Selected particles were refined with C3 symmetry by non-uniform refinement and subsequent local refinement was performed with a mask excluding micelles and disordered regions. Further 3D classification and local refinement were used to refine final map and local filtered map were used to build models.

For high concentration dataset, we extracted particles by a trained Topaz model and a consensus map was obtained by similar data processing procedure. The Topaz model was further trained from these particles to yield a refined consensus map. This consensus map represented weak density of TMD-bound density of ICD and weird density around TM5. Without any prior information, these particles were divided into closed and open states by ab initio reconstruction and heterogeneous refinement. Entire particles from 2D classification were classified by a series of heterogeneous refinement with these open and closed models, decoys and junks. Closed particles and closed-like particles (absence or weak ICD density) were further classified by heterogeneous refinement and refined by similar process to yield a closed state map. The rest open state particles were used to yield an open state map, but we obtained a low-resolution map due to heterogeneity of open state particles. Other constructs were also processed by similar data processing procedure. Detailed information of data processing is summarized in Supplementary Figs. 7 and 8. To produce an initial model, a homology model of AtSLAC1 was generated using Phyre2 web server[46] based on known HiTehA structures. The homology model was fitted into the open AtSLAC1 6D map using Phenix[47]. Coot[48] was used to manually rebuild models and real space refinement of Phenix was performed to deduce a final model. For model building of AtSLAC1 WT open, 6D closed and 8D closed, similar procedures were conducted, but the open AtSLAC1 6D and closed AtSLAC1 S59A model was used as an initial model. The final models contained residues 150-516 for AtSLAC1 6D closed structure, residues 182-318 and 329-506 for AtSLAC1 6D open structure, residues 182-317 and 327-508 for AtSLAC1 WT open structure and residues 105-112 and 148-516 for AtSLAC1 8D closed structure. The cryo-EM data collection, refinement and validation statistics of all structures are summarized in Supplementary Table 1. All models and maps in the figures were visualized using open-source PyMOL[49], UCSF Chimera[50] and UCSF ChimeraX[51], and were analyzed using MOLE[38] and PISA[52].

### Cell surface biotinylation and immunoblot analysis

To test cell surface expression level of AtSLAC1, surface biotinylation was performed. HEK293T cells (CRL-3216, ATCC) were transfected with wild-type and mutant constructs using the Transporter 5™ Transfection Reagent and incubated for 40 h. Cells were washed with ice-cold PBS and biotinylated using 2 mL of 0.25 mg mL$^{-1}$ (per 35 mL) EZ-Link Sulfo-NHS-SS-Biotin for 15 min on ice. Remaining reactive biotin was washed out with PBS and additionally quenched with 50 mM glycine pH 7.5.

Cells were harvested and lysed in a lysis buffer (1% Triton X-100, 1X PBS, 1X Protease inhibitor cocktail). Biotinylated plasma membrane (surface) proteins were purified from whole cell lysate (total) using NeutrAvidin Plus UltraLink resin. After washing the resin with the lysis buffer, proteins were eluted from resin by 2X LDS sample buffer. Total and surface proteins were separated on a Bolt™ 4–12% Bis-Tris Plus Gel (Invitrogen) and transferred to a PVDF membrane using iBlot 2 Transfer Stack. 250–500 ng antibodies [anti-HA (ab9110, Abcam), anti-Actin (8457 S, Cell Signaling Technology), and anti-transferrin receptor (TfR) (13208 S, Cell Signaling Technology)]/mL iBind Flex Solution were used for immunoblotting. Protein bands were visualized by Clarity Western ECL Substrate and images were acquired by the ChemiDOC imaging system. Three independent experiments were conducted. Densitometric analysis was performed by ImageJ[53] software.

## Whole-cell patch clamp recordings

HEK293T cells were maintained in Dulbecco's modified Eagle's medium (DMEM) supplemented with 10% fetal bovine serum (Thermo) at 37 °C in a 5% $CO_2$ incubator. Cells at 50–60% confluence in a 6-well plate were transiently transfected with 0.5 ~ 1.0 µg of AtSLAC1 plasmid DNA in complexed with polyethylenimine transfection reagent according to the manufacturer's instruction. For identifying transfected cells in whole-cell patch clamping, GFP-expressing plasmid DNA was co-transfected at 1:5 ratio. A day after transfection, the cells were moved to the poly-L-lysine (PLL) coated coverslips and allowed to be attached at least 6 h. Typically, electrophysiological recordings were conducted 30 ~ 48 h after transfection. Borosilicated micropipettes (WPI) were pulled by using P-1000 puller (Sutter) and fire-polished by using MF-900 microforge (Narishige). The patch pipettes had resistance of 2 ~ 3.5 MΩ were backfilled with the internal solution (140 mM CsCl, 20 mM HEPES, pH 7.4 with CsOH). The external solution contained 140 mM NaCl, 20 mM HEPES, pH 7.4 with NaOH. Whole-cell patch clamp recordings were performed with a Multiclamp 700B amplifier (Molecular Devices) controlled by pCLAMP 11 software via Digidata 1550B data acquisition system (Molecular Devices). After formation of whole-cell patch clamp mode, currents were evoked by 500-ms step stimulations of −100 mV to +100 mV with 20-mV increments from a holding potential of 0 mV. Data were sampled at 5 kHz and filtered at 2 kHz by low-pass vessel filter. Experiments were performed at room temperature (20-25 °C).

## Reporting summary

Further information on research design is available in the Nature Portfolio Reporting Summary linked to this article.

# Data availability

Coordinates have been deposited in the Protein Data Bank under accession codes 8GW6 (AtSLAC1 6D closed structure), 8GW7 (AtSLAC1 6D open structure), 8J0J (AtSLAC1 8D closed structure) and 8J1E (AtSLAC1 WT open structure). The corresponding cryo-EM density maps with local-filtering, half-maps and masks have been deposited in the Electron Microscopy Data Bank under accession codes EMDB-34303 (AtSLAC1 6D closed structure), EMDB-34304 (AtSLAC1 6D open structure), EMDB-35904 (AtSLAC1 8D closed structure) and EMDB-35920 (AtSLAC1 WT open structure). Previously published coordinates 7EN0 for BdSLAC1 and 7WNQ for AtSLAC1 S59A are available in the Protein Data Bank. Source data are provided with this paper.

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

## Acknowledgements

We thank Dr. Tae-Houn Kim at Duksung Women's University for discussion and staff members at Korea Basic Science Institute (KBSI, Leading Research Equipment User Program, C230430) for technical support. Computational and network resources were provided in part by KREONET and Global Science experimental Data hub Center (GSDC), Korea Institute of Science and Technology Information (KISTI). This work was supported by the Basic Science Research Program (2022R1A2C1009882), the Bio & Medical Technology Development Program (2021M3A9I4022936 and RS-2023-00223552) and the Science Research Center Program (2017R1A5A1014560) through the National Research Foundation of Korea (NRF) grants and a grant from the Next-Generation BioGreen 21 Program (PJ015672), Rural Development Administration, Republic of Korea to S.L., and by the Basic Research program through Korea Brain Research Institute (KBRI) funded by the Ministry of Science and ICT (22-BR-01-02) to H.-H.L.

## Author contributions

Y.L. prepared samples and performed structural analysis. H.S.J., E.J.D., K.K. and H.-H.L. performed patch-clamp analysis. S.J., C.T.H.L. and B.-G.K. prepared samples. J.H. performed expression level analysis. S.-H.J. performed cryo-EM analysis. Y.L., H.-H.L. and S.L. wrote the manuscript with help from all authors. S.L. directed and supervised all of the research.

## Competing interests

The authors declare no competing interests.
