## [Peer Review File · Nature Communications]

Cryo-EM structures of the plant anion channel SLAC1 from *Arabidopsis thaliana* suggest a combined activation modelReviewers' Comments:

Reviewer #1:

Remarks to the Author:

In this manuscript, Yeongmok and colleagues investigated the structure of SLAC1 6D mutant, an anion channel in Arabidopsis, using structural and electrophysiological approaches. The structure of phosphomimetic active mutant AtSLAC1 6D adopt distinct conformations of trimer, open and closed conformations. Based on structural analysis and functional assay, the authors proposed an combined activation mechanism that the interaction of ICL2 and ICD coupled to partial unwinding of TM5 and helical rearrangements of other TMs regulate conformational changes for channel opening. By mutational studies, the authors proposed ICL2 act as a switch that can sense the phosphorylation of SLAC1. Although this hypothesis requires further verification, it extend our understanding of working mechanism of SLAC1.

There are some concerns to be addressed to improve this manuscript:

Major points:

1. The reported WT SLAC1 structure as the authors mentioned (Li et al, Structure of the Arabidopsis guard cell anion channel SLAC1 suggests activation mechanism by phosphorylation. Nat Commun.), also identified five phosphorylation site (S59, S86, S120, S146 and S543), and proposed S59 and S86 are the most critical ones, which is not mentioned in this manuscript. In Extended Data Figure 1b, the current of phosphomimetic active mutant SLAC1 6D is smaller than WT. Does this indicate other undetected phosphorylation site exist? Which one or ones are the most important phosphorylation site and why? Meanwhile, it is recommended that the authors would explain why chose the SLAC1 6D construct at the very beginning although the reference has been cited.
2. The author proposed that ICL2 is a putative activation switch that can sense the phosphorylation of SLAC1 as its positively charged residue. Does 6D mutant have influences on ICL2 motif or the ancient ICL2 mutants? Will the ionic currents of EEALCA and VGRLCE mutants reduced much more than AtSLAC1 WT?

Minor points:

1. Extended Data Figure 1b-c: Are number of observations in Extended Data Figure 1b the same as Extended Data Figure 1c?
2. Figure 1f-g: Are number of observations in Figure 1f the same as Figure 1g?
3. Figure 2c. The ICL2-down state and the ICL3-down state is not clearly shown.
4. Line 503: # 20mM should be # 20 mM.
5. It is better to label the C3 symmetric refinement on the data processing workflow. A further concern is application of C3 symmetry refinement may average the densities. I wonder if the author can provide the result of C1 symmetry refinement to show the density of ICL2 is the same.
6. Figure 1c,d. It would be better to add a panel to present the structural differences in 3D models, as most of the TMs remains the same in the 2D-topological view.
7. The similar concern as point6, please indicate the ICL2 in 3D-model or Figure-1b.
8. Please show the side chains of KRRLCK motif on the closed state structure.

Reviewer #2:

Remarks to the Author:

General comment:

Activation of SLAC1 in guard cells via phosphorylation initiates fast stomatal closure in response to various signals such as ABA, high CO₂, SA, and hydroxy peroxide, limiting the evaporation of water from the leaf under unfavorable conditions. Recent studies determined the closed state structures of SLAC1 from the bacterial orthologue Haemophilus influenzae (HiTehA), the monocot plant *Brachypodium distachyon* (BdSLAC1) and the dicot plant *Arabidopsis thaliana* (AtSLAC1). Based on

these structures two alternative models have been proposed which are currently controversially discussed – the binding-activation model and the inhibition-release model (Deng et al 2021; Li et al 2022). Here, the authors resolved cryo-EM structures of a sextuple phosphomimetic AtSLAC1 mutant (called 6D) in open and closed states with a resolution of 3.3 Å. Comparison of those structures let the authors to suggest the structural basis for the opening of the conducting pore. Thereby phosphorylation of an intracellular domain (ICD) leads to its dissociation from TMDs. A positively charged intracellular loop ICL2 between TMD4 and 5 is suggested to sense the negatively charged phosphorylated intracellular domain. In response to the interaction between ICL2 and ICD, conformational changes and a partial unwinding of TMD5 is accompanied by the widening of the channel's gate which is believed to open the conducting ion pore. In addition to the pore opening mechanism, several putative anion binding sites were uncovered. Thus, the authors suggest that SLAC1 anion channels operate a mechanism combining binding-activation and inhibition-release models.

The MS is well-written, the figures of high quality, and the topic of the paper is of major interest for structural biology and membrane transport field in particular and the broad readership of Nature Communications in general. Bevor referees can recommend accepting this paper for publication after the listed major comments had been properly addressed.

Major comments

- 1) SLAC1 F450 and T513D The authors found the group of transporters based on the homology to bacterial and human Magnesium efflux transporters. Biased by these homologies the experiments were designed with the aim to find a Magnesium related phenotype. In the light of this biased approach the referee requests a functional prove of the Magnesium transport function, its voltage dependency and specificity in a heterologous expression system other than the Magnesium transport deficient MM281 cells. The MGR transporters/facilitators should be electrogenic, thus expression in *X. oocytes* or mammalian/insect cell lines should give rise to ionic currents.
- 2) Lines 169-196: The authors replaced the positively charged AtSLAC1 ICL2 by neutral or negatively charged ICL2s of MpSLAC1 and KnSLAC1 to prove functional relevance of the ICL2. However, the electrophysiological experiments in HEK cells did not give the expected results – there was no significant difference between AtSLAC1 6D and the mutants carrying the ICL2s of the ancient SLAC1s. The authors argue that the absence of two arginine residues (R321 and R322) in ICL2 might be required for structural integrity of the closed conformation. To the referee's opinion, the authors should replace the ancient amino acids residues by RR in the ICL2 domain to prove their conclusion/suggestion. Even with these two positive charges in the ancient ICL2 domains the AtSLAC1 mutants have fewer positive charges than the AtSLAC1 6D. Currently, this paragraph does not provide any additional information/conclusion for the reader and the statement: "Taken together, we propose that ICL2 is a putative activation switch that can sense the phosphorylation of SLAC1" is not justified by the electrophysiological data.
- 3) Related to point 2): The authors should consider using *Xenopus oocytes* for their biophysical analysis of AtSLAC1 and the respective mutants thereof. Why? In *oocytes* AtSLAC1 WT is electrically silent and is only active in the presence of plant-derived SLAC1-activating kinases. This is much closer to the physiological conditions than in HEK cells, where an unknown kinase activates AtSLAC1 WT at phosphorylation sites that are unknown so far.
- 4) A major issue of the MS is the "Anion conductance pathway in the open and closed states" and the "Conformational changes of the central gate residues by helical rearrangement". E.g., the authors conducted pore analysis of the open and closed structures using MOLE 2.5 and propose mechanisms how the pore structure changes from the closed to the open state that seems to involve unwinding of TM5, bending and movements of TMs, the branching from P1 in a second pore (P2) as well as changes of pore lining residues. In this context, the referee wonders why single mutations of the central gate residue F450 is sufficient to fully open the ion pathway of SLAC1-type anion channels without the need of a phosphorylation (c.F. Chen et al 2010; Lind et al 2013). Moreover, the T513D mutant in the C-terminus of AtSLAC1 exhibited kinase-independent SLAC1 activity, too (Maierhofer et al 2014). Although the authors discussed the F450A mutant, Cryo-EM structures of these mutants (F450A,

T513D) compared to the two states of the AtSLAC1 6D mutant presented here would further deepen our understanding of the activation mechanism of SLAC1.

5) As pointed out by the authors, Cryo-EM structures in general do not consider the membrane potential and its influence on the activation state and thus conformation of the protein of interest. The authors state in lines 424 to 425: "Since our cryo-EM structures were obtained in the absence of applied voltage, our open structure is likely to reflect a non- or weakly conductive state of AtSLAC1 at zero voltage". This is not in line with the electrophysiological dataset shown in Figure 1f and Extended Data Figure 1 where the currents did not exhibit any voltage dependence. Moreover, other publication with AtSALC1 expressed in oocytes (e.g. Geiger et al., 2009) or HEK cells (e.g. Li, Y. et al. 2022) or AtSLAC1 currents recorded from guard cell protoplasts (e.g. Geiger et al 2009) show no or only a weak voltage dependence. If at all, SLAC1 is more open at 0 mV than at hyperpolarized membrane potentials (e.g. see open probabilities determined by Maierhofer et al 2014; Geiger et al 2009). Thus, the argumentation of the authors is inconsistent to their own electrophysiological data (Figure 1f and Extended Data Figure 1) and the depolarization-dependent activation of AtSLAC1 observed by others.

Minor comments

1) Line 70 to 72: The sentence is not clear "The inhibition-release model states that the binding of ICDs to TMD renders SLAC1 to assume a self-inhibitory state and phosphorylation of serine/threonine residues of ICDs relieves such inhibition, thereby activating SLAC1. "

2) Line 115 and 116: "Since AtSLAC1 WT 115 showed more heterogenous conformational states than AtSLAC1 6D" the reader may ask why these different conformation were not analyzed?

3) 123 to 132: the reader might ask, why did the authors select these two datasets and did not an unbiased approach analyzing the particles from all grid preparation conditions?

4) Line 274: "but movement of F266 and S447 during open" should read "but movement of F266 and S447 during opening"

5) Line 421 and 422: redundant sentences "We presume that our open structure of AtSLAC1 6D represents a low conductive open state at zero voltage. AtSLAC1 represents low conductivity without voltage".

6) Line 444: „However, when it phosphorylated, F266A..."delete "it".

Point-by-point response

We appreciate the constructive comments by the reviewers. Below we provide our responses to the issues raised by the reviewers.

Reviewer #1

1. The reported WT SLAC1 structure, as the authors mentioned [Li et al, Structure of the Arabidopsis guard cell anion channel SLAC1 suggests activation mechanism by phosphorylation. Nat Commun, 2022], also identified five phosphorylation sites (S59, S86, S120, S146 and S543), and proposed S59 and S86 are the most critical ones, which is not mentioned in this manuscript.

Pursuant to the reviewer's comment, we included the references proposing S59 and S86 as critical phosphorylation sites. We revised our manuscript as follows to reflect this point:

[Lines 111-112 of the revised manuscript]

“Recent studies have suggested that two phosphorylation sites, S59 and S86, are crucial for SLAC1 activation^{28,31}.”

Meanwhile, it is recommended that the authors would explain why chose the SLAC1 6D construct at the very beginning although the reference has been cited.

When we attempted to determine the structure of SLAC1, there was no available SLAC1 structure such as BdSLAC1 and AtSLAC1 S59A structures. When we planned to study an active state conformation of SLAC1, we assumed that recombinant WT SLAC1 might have heterogeneous phosphorylation states with multiple conformations, which would render structure determination very challenging even with cryo-EM technology. Therefore, we opted to search from literature a phosphomimetic mutant that would recapitulate an active state of SLAC1 with a homogeneous phosphorylation state. We chose a reported phosphomimetic active mutant termed as AtSLAC1 6D in this manuscript for structure determination by cryo-EM. We later realized that AtSLAC1 WT represents single open state by cryo-EM.

In Supplementary Figure 1b, the current of phosphomimetic active mutant SLAC1 6D is smaller than WT. Does this indicate other undetected phosphorylation site exist? Which one or ones are the most important phosphorylation site and why?

Since we observed an open conformation in 6D mutant by cryo-EM, we investigated the electrophysiological and structural effects of phosphorylation on S59 and S86 by introducing the aforementioned mutations to our AtSLAC1 6D backbone, producing 7D (6D + S59D) and 8D (6D + S59D/S86D). Unlike the previous study [Li et al, Nat Commun, 2022], the introduction of both or either of phosphomimetic S59D or S86D mutation to the 6D mutant caused a significant decrease of current density to the level comparable with that of the phospho-defective S59A mutant (Figure 1a and Supplementary Figure 1e-g). When we tested either phosphomimetic S59D and S59E single mutant, both mutants showed impaired conductivity whose levels were compatible with that of S59A mutant (Supplementary Figure 1f, g). Cryo-EM maps of AtSLAC1 7D and 8D revealed that the two mutants assume the closed conformation at the concentrations similar to that of AtSLAC1 6D open structure. Electrophysiological data showed reduced currents, corroborating the structural observation of the closed structures of both 7D and 8D (Figure 1a and Supplementary Figure 1e-g). These results suggest that phosphomimetic aspartate mutations on two major phosphorylation sites, S59 and S86, poorly mimic natural phosphorylation. Nevertheless, AtSLAC1 WT and its phosphomimetic mutants with various conductive activities enabled us to find out conformational changes according to their functionality (Figure 1a).

The cryo-EM structural analysis revealed an inhibitory component in addition to the “plug”, which we termed as “latch” (Supplementary Figure 10 and Figure 5a). Such latch was observed in all the cryo-EM maps in the closed state. AtSLAC1 8D has more clear density of latch than 6D and S59A (PDB: 7WNQ). Based on structure and multiple sequence alignment, we assumed that the latch is regulated by the patch 1 [Deng et al, PNAS, 2021] spanning four phosphorylation sites (S113, T114, S116 and S120) (Figure 5a). To investigate the effects of phosphorylation on the sites in the patch 1, we prepared two phosphomimetic mutants termed 10D and 12D that incorporated S113D/T114D/S116D/S120D to the 6D and 8D mutant, respectively. The 12D mutant restored current densities that were impaired by the 8D mutant (Figure 5b, c), which is clearly distinct from the impaired current densities by aspartate mutations on major sites. In contrast, the 10D mutant, which was expected to have weak latch binding due to faint density of the latch, exhibited current densities similar to those by the 6D mutant, indicating no effect of the patch 1 on the ICL2-mediated activation of SLAC1. Taken together, we suggest that each phosphorylation site may have different roles and mechanism in activation. We revised our manuscript to reflect these points as follows:

[Lines 150-156 of the revised manuscript]

“We also determined the open state of AtSLAC1 WT and the closed state of AtSLAC1 8D with C3 symmetry, at resolution of 3.8 Å and 2.7 Å, respectively (Supplementary Fig. 4, 5, 7). Both closed and open states are nearly identical to those of 6D. C α atoms root mean square deviation (C α r.m.s.d.) between the open states was 0.58 Å, and that of closed states was 0.42 Å. The AtSLAC1 7D showed a low resolution (4.37 Å) and was not modeled due to poor grid state including ice contamination, but it also represented the closed state that is identical to others.”

[Lines 170-197 of the revised manuscript]

“We found the existence of an unidentified density (hereinafter called as “latch”) nearby the plug region in the cryo-EM maps of the closed structures of AtSLAC1: 6D and 8D as observed in S59A (PDB: 7WNQ)³¹ (Fig. 1d, Supplementary Fig. 10a). However, ambiguous and disconnected densities in the maps of AtSLAC1 6D and S59A³¹ made it difficult to discriminate them as protein densities (Supplementary Fig. 10b). By contrast, the cryo-EM map of AtSLAC1 8D exhibited a clear density for the latch with the sequence of ¹⁰⁵DF(S107D)MFRTK¹¹², revealing features for unequivocal modeling: apparent N-terminal direction due to right-helical turn and side chain orientation; readily discernible densities of the unique amino acid pattern (F/Y)xx(F/Y)R; distinct density of phosphomimetic mutation S107D compared to that of the corresponding S107 in the AtSLAC1 S59A³¹ (Supplementary Fig. 10b). Additionally, S107D and S152D phosphomimetic mutations were visible from latch and plug in our AtSLAC1 6D and 8D structures, but these mutations caused negligible structural differences compared to AtSLAC1 S59A³¹ (Supplementary Fig. 10b, c). The latch is located nearby the plug region, but the gap between the C-terminal end of the latch (K112) and the N-terminal end of the plug (N148) is long with a length of 35 amino acids (Supplementary Fig. 10c). The latch binds to both TMD and the plug with a buried surface area (BSA) of 522.7 Å². The binding of latch is expected to stabilize the plug by increasing interface between TMD and ICD (plug and latch) from 1176.8 Å² to 1446.9 Å² BSA (Supplementary Fig. 10d). To confirm whether the latch and plug are general features of SLAC1 or not, we conducted multiple sequence alignments of orthologous SLAC1 sequences from dicot, monocot and basal plants and two ancient SLAC1 orthologues from a charyophyte *Klebsormidium nitens* (Kn) and a liverwort *Marchantia polymorpha* (Mp) based on a previous study³⁴ (Supplementary Fig. 11). Both KnSLAC1 and MpSLAC1 are OST1-mediated, phosphorylation-insensitive SLAC1 due to lack of OST1-specific phosphorylation sites³⁵. All plant SLAC1s except two ancient SLAC1 orthologues have well conserved N- and C-terminal plug (Supplementary Fig. 11). By contrast, the latch is highly

conserved in dicot and monocot, but less conserved in basal plants (Supplementary Fig. 11). These results suggest that the plug is a common inhibitory component of SLAC1 in all plants, and that the latch is an additional inhibitory component in higher plants.”

[Lines 425-443 of the revised manuscript]

“Recent studies suggest that SLAC1 harbors two major phosphorylation sites S59 and S86, and two patches (patch 1 and 2) that are crowded with minor phosphorylation sites^{28,31}. Our structures uncover that distribution of phosphorylation sites in the intracellular space is restricted by spatial constraints from the ordered plug and latch (Fig. 5a). The phosphorylation motif of OST1 (¹¹⁰RxxS¹¹³)⁴⁰ spans the latch (residues 105-112) and the patch 1 (S113, T114, S116 and S120) (Fig. 5a, Supplementary Fig. 11). The latch, patch 1 and OST1 phosphorylation motif are well conserved throughout dicot and monocot, and also found in some basal plants (Supplementary Fig. 11). We envisage that phosphorylation of the patch 1 affects binding of the latch or vice versa. To verify whether phosphorylation on the patch 1 releases the inhibition from the latch, we introduced the phosphomimetic mutations (S113D/T114D/S116D/S120D) of the patch 1 to the AtSLAC1 6D and 8D mutant that have the well-bound latch from cryo-EM data (Supplementary Fig. 10b), yielding 10D and 12D mutants, respectively. Inevitably, the phosphomimetic mutations on the patch 1 residues of the 8D mutant (“12D”) restored current densities compared to the 8D mutant (Fig. 5b, c). In contrast, the same phosphomimetic mutations on the patch1 residues of the 6D mutant (“10D”) did not make a difference, reflecting their weakly-bound latch from cryo-EM (Fig. 5b, c, Supplementary Fig. 10b). Unlike defective aspartate mutations on the major sites, aspartate mutations in the patch 1 induced activation of SLAC1. These results suggest that various phosphorylation sites have different roles in the activation of SLAC1.”

2. The author proposed that ICL2 is a putative activation switch that can sense the phosphorylation of SLAC1 as its positively charged residue. Does 6D mutant have influences on ICL2 motif or the ancient ICL2 mutants? Will the ionic currents of EEALCA and VGRLCE mutants reduced much more than AtSLAC1 WT?

First of all, we apologize for the error in describing the consensus ICL2 sequence of charophytes including KnSLAC1: it should have been EERxxA, not EEAxxA (Figure 1f). We now corrected such an error in the revised manuscript.

Pursuant to the reviewer's suggestion, we tested both ancient ICL2 motifs and various ICL2 mutants, XXXLCX (X = A, N, D, E) as well as WT ICL2 motif of AtSLAC1 (Figure 1g, h and Supplementary Figure 12). The two out of 4 ICL2 mutants, VGRLCE and NNNLCN, in the WT background exhibited statistically significant reduction in the current densities (Supplementary Figure 12c). By contrast, the no ICL2 mutant in the 6D mutant background showed statistically significant change (either increase or decrease) in the current densities. These results are in accordance with poor mimicry of aspartate mutation on some phosphorylation sites. Furthermore, we confirmed that two arginine residues on ICL2 are also important for formation of closed state, not only for activation as assumed in previous manuscript. We described detailed results about ICL2 mutants in revised manuscript as follows:

[Lines 232-255 of the revised manuscript]

“To verify the functional relevance of ICL2 on channel activation, we mutated these positively charged residues (KRRxxK) of AtSLAC1 to corresponding residues of phosphor-insensitive KnSLAC1 (EERxxA), MpSLAC1 (VGRxxE) and various amino acid substitutions (AAAxxA, NNNxxN, DDDxxD and EEExxE) (Fig. 1g, h). AtSLAC1 WT (³²⁰KRRLCK³²⁵) and its ICL2 mutants were expressed in HEK293T cells, and whole-cell currents from these constructs were recorded by whole-cell patch clamp recordings. Compared to WT AtSLAC1, the ionic currents of most ICL2 mutants showed significantly reduced current density as that of phospho-defective S59A mutant, but EERLCA (KnSLAC1) and AAALCA (alanine substitution) mutants showed less reduced current density (Fig. 1g, h). Two arginine residues in ICL2, R321 and R322, are expected to be required for structural integrity of the closed structure, unlike two lysine residues K320 and K325 that are exposed to the intracellular space (Supplementary Fig. 12a). The two arginine residues interact with neighboring residues: V151, F259, Q314 and S317 in the same protomer; L372 and Q374 on adjacent protomer in the closed structures of AtSLAC1 6D, 8D and S59A (PDB: 7WNQ) (Supplementary Fig. 12a). We restored these two arginine residues from above mutants to exclude possibility that reduced current density might have come from structural collapse. The results showed that mutations on two solvent-exposed lysine residues are sufficient to abolish the activation of SLAC1 (Supplementary Fig. 12b). Also, restoration of two arginine residues from moderate conductive EERLCA and AAALCA mutants resulted in a much lower conductivity, implicating that the two arginine residues of ICL2 is important for maintenance of closed conformation (Fig. 1g, h, Supplementary Fig. 12b).

In contrast, ICL2 mutants on AtSLAC1 6D showed non-significant decrease of current density (Supplementary Fig. 12c). It seemed to reflect weak interactions between ICD and ICL2 in 6D mutant. Taken together, these results demonstrate that ICL2 is a putative activation switch that can sense the phosphorylation of SLAC1.”

Minor points

1. Supplementary Figure 1b-c: Are number of observations in Supplementary Figure 1b the same as Supplementary Figure 1c?

Those two figures came from the same observations. We revised the corresponding figures and legends accordingly.

2. Figure 1f-g: Are number of observations in Figure 1f the same as Figure 1g?

Those two figures came from the same observations. We revised the corresponding figures and legends accordingly.

3. Figure 2c. The ICL2-down state and the ICL3-down state is not clearly shown.

We changed the terms “up state” and “down state” to “extended state” and “bent state” for clarity. We also added Figure 2a and 2d that clearly showed the extended and bent states of ICL2 and ICL3.

4. Line 503: # 20mM should be # 20 mM.

The erratum has been corrected.

5. It is better to label the C3 symmetric refinement on the data processing workflow. A further concern is application of C3 symmetry refinement may average the densities. I wonder if the author can provide the result of C1 symmetry refinement to show the density of ICL2 is the same.

We revised workflow figures to label the symmetry and added panels showing refinement results of C1 symmetry for WT and 6D open structures (Supplementary Figure 9c-d). The results of C1 symmetry refinements of the open state WT and 6D showed asymmetric shapes of ICD and ICL2, but symmetric shapes of TMD. We observed that ICL2s were extended from the center of the SLAC1 trimer and connected to ICD regardless of the symmetry applied (C1 or C3). We revised our manuscript to reflect these points as follows:

[Lines 209-214 of the revised manuscript]

“To exclude artificial features from enforced C3 symmetry, we also conducted refinement without symmetry in both AtSLAC1 WT and 6D open state (Supplementary Fig. 9c, d). Despite no applied symmetry, all protomers of WT and 6D open state are almost identical except ICL2 and ICD that have asymmetrical distribution as expected from weak density with C3 symmetry (Supplementary Fig. 9c, d). Nevertheless, ICL2s are extended from the center of trimer, and connected to ICD regardless of applied symmetry (Supplementary Fig. 9b, c, d).”

6. Figure 1c, d. It would be better to add a panel to present the structural differences in 3D models, as most of the TMs remains the same in the 2D-topological view.

We revised Figures 1, 2 and other figures so that it can emphasize the structural differences between closed and open states.

7. The similar concern as point 6, please indicate the ICL2 in 3D-model or Figure-1b.

To resolve the concern about equivocal representation about ICL2 movement, we indicated the ICL2 in Figure 1d-e and Figure 2. For clarity, we also colored ICL2 and ICL3 of almost figures as blue and red, respectively.

8. Please show the side chains of KRRLCK motif on the closed state structure.

We added a panel to represent the side chains of ICL2 residues (Supplementary Figure 12a) and revised Figures showing the side chains of positively charged residues of ICL2 (Figures 2, 4, Supplementary Figure 12).

Reviewer #2

1) SLAC1 F450 and T513D: The authors found the group of transporters based on the homology to bacterial and human Magnesium efflux transporters. Biased by these homologies the experiments were designed with the aim to find a Magnesium related phenotype. In the light of this biased approach the referee requests a functional prove of the Magnesium transport function, its voltage dependency and specificity in a heterologous expression system other than the Magnesium transport deficient MM281 cells. The MGR transporters/facilitators should be electrogenic, thus expression in X. oocytes or mammalian/insect cell lines should give rise to ionic currents.

We believe that this issue has nothing to do with SLAC1 because SLAC1 is an anion channel and not a magnesium efflux transporter. It seems that this issue was unintentionally misplaced.

2) Lines 169-196: The authors replaced the positively charged AtSLAC1 ICL2 by neutral or negatively charged ICL2s of MpSLAC1 and KnSLAC1 to prove functional relevance of the ICL2. However, the electrophysiological experiments in HEK cells did not give the expected results – there was no significant difference between AtSLAC1 6D and the mutants carrying the ICL2s of the ancient SLAC1s. First of all, we apologize for the error in describing the consensus ICL2 sequence of charophytes including KnSLAC1: it should have been EERxxA, not EEAxxA (Figure 1f). We now corrected such an error in the revised manuscript.

The ICL2 mutants in WT showed statistically significant differences in current densities compared to those by the ICL2 mutants in 6D (Supplementary Figure 12c). We assumed that less significant differences of 6D came from weak interactions between ICD and ICL2, which is consistent with the insufficient activities of phosphomimetic aspartate mutations.

The authors argue that the absence of two arginine residues (R321 and R322) in ICL2 might be required for structural integrity of the closed conformation. To the referee’s opinion, the authors should replace the ancient amino acids residues by RR in the ICL2 domain to prove their conclusion/suggestion. Even with these two positive charges in the ancient ICL2 domains the AtSLAC1 mutants have fewer positive charges than the AtSLAC1 6D. Currently, this paragraph does not provide any additional information/conclusion for the reader and the statement: “Taken together, we propose that ICL2 is a putative activation switch that can sense the phosphorylation of SLAC1” is not justified by the electrophysiological data.

Pursuant to the reviewer’s suggestion, we tested RR restoration mutants in the background of WT (Supplementary Figure 12b). RR restorations (1) from ancient EERLCA (charophyte) to ERRLCA and (2) from a sort of “bare” ICL2 motif mutant AAALCA to ARRLCA decreased their current densities as expected (Supplementary Figure 12b). These results suggest that the two arginine residues also have a role in closed conformation maintenance. We revised our manuscript to reflect these points as follows:

[Lines 232-255 of the revised manuscript]

“To verify the functional relevance of ICL2 on channel activation, we mutated these positively

charged residues (KRRxxK) of AtSLAC1 to corresponding residues of phosphor-insensitive KnSLAC1 (EERxxA), MpSLAC1 (VGRxxE) and various amino acid substitutions (AAAxxA, NNNxxN, DDDxxD and EEExxE) (Fig. 1g, h). AtSLAC1 WT (³²⁰KRRLCK³²⁵) and its ICL2 mutants were expressed in HEK293T cells, and whole-cell currents from these constructs were recorded by whole-cell patch clamp recordings. Compared to WT AtSLAC1, the ionic currents of most ICL2 mutants showed significantly reduced current density as that of phospho-defective S59A mutant, but EERLCA (KnSLAC1) and AAALCA (alanine substitution) mutants showed less reduced current density (Fig. 1g, h). Two arginine residues in ICL2, R321 and R322, are expected to be required for structural integrity of the closed structure, unlike two lysine residues K320 and K325 that are exposed to the intracellular space (Supplementary Fig. 12a). The two arginine residues interact with neighboring residues: V151, F259, Q314 and S317 in the same protomer; L372 and Q374 on adjacent protomer in the closed structures of AtSLAC1 6D, 8D and S59A (PDB: 7WNQ) (Supplementary Fig. 12a). We restored these two arginine residues from above mutants to exclude possibility that reduced current density might have come from structural collapse. The results showed that mutations on two solvent-exposed lysine residues are sufficient to abolish the activation of SLAC1 (Supplementary Fig. 12b). Also, restoration of two arginine residues from moderate conductive EERLCA and AAALCA mutants resulted in a much lower conductivity, implicating that the two arginine residues of ICL2 is important for maintenance of closed conformation (Fig. 1g, h, Supplementary Fig. 12b). In contrast, ICL2 mutants on AtSLAC1 6D showed non-significant decrease of current density (Supplementary Fig. 12c). It seemed to reflect weak interactions between ICD and ICL2 in 6D mutant. Taken together, these results demonstrate that ICL2 is a putative activation switch that can sense the phosphorylation of SLAC1.”

3) Related to point 2): The authors should consider using Xenopus oocytes for their biophysical analysis of AtSLAC1 and the respective mutants thereof. Why? In oocytes AtSLAC1 WT is electrically silent and is only active in the presence of plant-derived SLAC1-activating kinases. This is much closer to the physiological conditions than in HEK cells, where an unknown kinase activates AtSLAC1 WT at phosphorylation sites that are unknown so far.

We used HEK cells for biophysical analysis because the HEK cell system has been successfully used to recapitulate the property of SLAC1 [Li et al, Nat Commun, 2022]. To minimize the possibility of interference of SLAC1 function by unknown kinases as mentioned by the reviewer, we performed

control experiments to ensure that SLAC1 WT and S59A behave as reported elsewhere [Li et al, Nat Commun, 2022] (Figure 1a, Supplementary Figure 12b). Based on these results, we believe that the HEK cell system would be sufficient to test electrophysiological properties of SLAC1 WT and mutants.

4) A major issue of the MS is the “Anion conductance pathway in the open and closed states” and the “Conformational changes of the central gate residues by helical rearrangement”. E.g., the authors conducted pore analysis of the open and closed structures using MOLE 2.5 and propose mechanisms how the pore structure changes from the closed to the open state that seems to involve unwinding of TM5, bending and movements of TMs, the branching from P1 in a second pore (P2) as well as changes of pore lining residues. In this context, the referee wonders why single mutations of the central gate residue F450 is sufficient to fully open the ion pathway of SLAC1-type anion channels without the need of a phosphorylation (c.F. Chen et al 2010; Lind et al 2013).

To obtain structural insights on the F450A mutant, we carefully examined the previous studies. The cryo-EM structures of HiTehA mutants – F262A, V and L (corresponding to AtSLAC1 F450) and G15D (corresponding to AtSLAC1 G194) – revealed that the structures of these mutants were almost identical to that of WT, leading the authors to assume that these mutants only loosen or block the pore [Chen et al 2010]. Functionally, F450A (or equivalent mutant) mutant showed higher conductivity and were not activated by OST1 phosphorylation [Chen et al 2010; Lind et al, 2015; and Deng et al, 2021]. The AtSLAC1 G194D mutant (or equivalent mutant) showed impaired current similar to that of HiTehA G15D due to pore blockage. These results support the idea that AtSLAC1 and HiTehA have similar conductance pathway. However, there are differences between plant SLAC1 and HiTehA. Plant SLAC1s have a narrow intracellular pore region compared to a wide pore of HiTehA. In HiTehA, F262 is exposed to both intracellular and extracellular side solvents and G194 is also exposed to solvents, thereby whole structures are less affected by mutation of these residues. In contrast, F450 and G194 of AtSLAC1 in the closed state are surrounded by hydrophobic residues, thereby F450A expected to create hydrophobic cavity and G194D cause steric hindrance to nearby pore residues. While open, extension of TM5 make cavity, followed by movement of TM7 and other TMs. The deformable intracellular side of SLAC1 could be affected by mutations that make cavity or steric clash. Also, abolished phosphorylation-dependent activation of F450A suggests that the mutation causes structural changes differing from that of WT.

Moreover, the T513D mutant in the C-terminus of AtSLAC1 exhibited kinase-independent SLAC1

activity, too (Maierhofer et al 2014).

Pursuant to the reviewer's point, we added a panel showing that T513, along with other phosphorylation sites such as S107D and S152D, is located in the ordered plug and latch (Supplementary Figure 10c). Specifically, T513 is positioned in the C-terminal plug region and juxtaposed to several charged residues R155, E172, D173, K514, R515, and K516 on the N-terminal plug region as described in previous study [Li et al, Nat Commun, 2022]. We envisage T513D mutant would impair the plug formation, leading to increased conductance.

Although the authors discussed the F450A mutant, cryo-EM structures of these mutants (F450A, T513D) compared to the two states of the AtSLAC1 6D mutant presented here would further deepen our understanding of the activation mechanism of SLAC1.

As the reviewer pointed out, structure determination of these mutants would deepen the understanding of the SLAC1 activation mechanism. However, F450A structure would be expected to exhibit minor conformational changes based on the literature search as discussed above. T513D structure is unlikely to yield a structure with clear local conformational changes based on our structures (Figure 5a). Therefore, we focused on collecting cryo-EM data for SLAC1 7D and 8D mutants. We would be delighted to pursue investigation on analyzing the structural consequences of the aforementioned mutants (F450A and T513D) in follow-up studies. We incorporated our responses to this issue in the revised manuscript as follows:

[Lines 460-461 of the revised manuscript]

“Phosphorylation sites are located on the disordered region of ICD except S107, S152 and T513 on the plug^{28,29,31} (Supplementary Fig. 10c, 11).”

[Lines 503-523 of the revised manuscript]

“Our results corroborate emerging roles of the pore-blocking phenylalanine residues. The bacterial homologue HiTehA structure has the gate residue F262 (corresponding to AtSLAC1 F450) that blocks wide pore³². F262A showed higher conductivity, but structure was almost identical to HiTehA WT. Several studies confirmed that equivalent F450A or corresponding mutation represented higher conductivity similar to that of bacterial homologue HiTehA^{28,32,35}. Therefore, the F450 of plant SLAC1 has simply been regarded as a pore blocking residue, like its ancestor. However, recent plant SLAC1 structures represented that intracellular side of pore

is narrowed by pore residues including bulky phenylalanine residues F266, F276 and F450^{28,31}. Either F276A or F450A mutant showed increased conductance without phosphorylation, presumably due to pore dilation from closed state²⁸. By contrast, when phosphorylated, F266A, F276A or F450A mutant features decreased conductance despite of the pore dilation^{28,31}. Thus, further conformational changes upon opening that differ from closed state were expected^{28,31}. These are consistent with conformational changes that we uncovered during closed to open transition. F266 has a crucial role for TM3 extension by sterically blocking TM7, followed by the central gate changes (Fig. 4c-e). F266A may lead to insufficient steric hindrance to induce helical extension of TM3. The less steric hindrance by F266A is apparently linked to the impairment of the central gate opening by F276 rotation (Fig. 4d, e). F276A and F450A mutation is also expected to cause structural impairment. F276 and F450 are surrounded by hydrophobic residues. SLAC1 has deformable intracellular side that allow helical rearrangement. Cavity derived from F276A and F450A mutations may induce unexpected structural changes differing from that of open state.”

5) As pointed out by the authors, cryo-EM structures in general do not consider the membrane potential and its influence on the activation state and thus conformation of the protein of interest. The authors state in lines 424 to 425: “Since our cryo-EM structures were obtained in the absence of applied voltage, our open structure is likely to reflect a non- or weakly conductive state of AtSLAC1 at zero voltage”. This is not in line with the electrophysiological dataset shown in Figure 1f and Supplementary Figure 1 where the currents did not exhibit any voltage dependence. Moreover, other publication with AtSLAC1 expressed in oocytes (e.g. Geiger et al., 2009) or HEK cells (e.g. Li, Y. et al. 2022) or AtSLAC1 currents recorded from guard cell protoplasts (e.g. Geiger et al 2009) show no or only a weak voltage dependence. If at all, SLAC1 is more open at 0 mV than at hyperpolarized membrane potentials (e.g. see open probabilities determined by Maierhofer et al 2014; Geiger et al 2009). Thus, the argumentation of the authors is inconsistent to their own electrophysiological data (Figure 1f and Supplementary Figure 1) and the depolarization-dependent activation of AtSLAC1 observed by others. We apologize for the erroneous opinions on voltage dependence. As the reviewer pointed out, voltage has negligible influence in activation of SLAC1. Pursuant to the reviewer’s point, we removed the paragraphs discussing voltage dependence in the revised manuscript.

Minor points

1) Line 70 to 72: The sentence is not clear “The inhibition-release model states that the binding of ICDs to TMD renders SLAC1 to assume a self-inhibitory state and phosphorylation of serine/threonine residues of ICDs relieves such inhibition, thereby activating SLAC1.”

Pursuant to the reviewer’s comment, we paraphrased the sentence to understand readily as follows:

[Lines 69-71 of the revised manuscript]

“The inhibition-release model states that ICDs work as self-inhibitory plugs that block the pore and phosphorylation of serine/threonine residues of ICDs relieves such inhibition, thereby activating SLAC1.”

2) Line 115 and 116: “Since AtSLAC1 WT 115 showed more heterogenous conformational states than AtSLAC1 6D” the reader may ask why these different conformations were not analyzed?

That sentences can be misleading, so we removed it. The cryo-EM data of AtSLAC1 WT showed a single open state that is identical with open state 6D, except more dispersed density of ICL2 and ICD. Additionally, we added sentences that explained about each construct, and incorporated cryo-EM maps of both WT and 6D without applied symmetry in Supplementary Figure 9.

3) Line 123 to 132: the reader might ask, why did the authors select these two datasets and did not an unbiased approach analyzing the particles from all grid preparation conditions?

Currently, we only have limited time for using cryo-EM facility, thereby most datasets were collected with different time, facility and parameters. So, it is difficult to conduct unbiased approach by merging all datasets.

4) Line 274: “but movement of F266 and S447 during open” should read “but movement of F266 and S447 during opening”

The corresponding sentence has been revised accordingly.

5) Line 421 and 422: redundant sentences “We presume that our open structure of AtSLAC1 6D represents a low conductive open state at zero voltage. AtSLAC1 represents low conductivity without voltage.”

That sentences were deleted in response to the major issue #5.

6) Line 444: „However, when it phosphorylated, F266A...” delete “it”.

The error has been corrected in the revised manuscript.

Reviewers' Comments:

Reviewer #1:

Remarks to the Author:

In the revised manuscript, the authors have addressed the previous points effectively and have provided additional electrophysiological data that adequately addresses the function of ICL2. Additionally, the authors have included comparisons between the wild-type (WT) in an open state, 8D in a closed state, and 6D in both open and closed states. Furthermore, the authors have identified a new element called the "latch" and have discussed the significance of a stable closed conformation. In order to enhance the interpretation of SLAC1 conformational changes, we propose the following perspectives for further discussion:

1. Line 117: The authors suggest that the significant decrease in current density observed with the S59D or S86D mutations in the 6D mutant may be due to poor natural phosphorylation mimicry. To further support this viewpoint, could co-expression with OST1 be considered based on the 6D mutant?

2. Line 197: The authors introduce the concept of the latch as an additional inhibitory component in higher plants. It would be interesting to investigate its existence and verify it through electrophysiology experiments. For example, deletion or replacement of a specific segment within 105-112 could be explored.

3. Line 210: By comparing the consistently extended ICL2 in WT SLAC1 with 6D SLAC1 in the open state, the authors demonstrate that ICL2 is a prominent feature. It would be valuable to improve the resolution of ICL2 in order to confirm the density in this region (Supplementary Fig. 9b, c, d) and determine if there is an interaction between ICD and ICL2. Additionally, the authors could discuss why SLAC1 alone maintains an open conformation in the absence of kinase and whether this conflicts with the activation mechanism in stomata (Geiger D et al 2009; Maierhofer T et al 2014).

4. Line 219: The authors conclude that the interaction between ICL2 and ICD is likely to induce conformational changes, but there appears to be a lack of evidence in the structure. Is it possible that the movement of the latch and plug creates space by relieving steric hindrances, thereby allowing ICL2 to move downwards? Instead of focusing solely on the conformational change associated with the release of inhibition, it may be informative to compare the structural differences in ICL2 between WT and NNNLCN to confirm that the loosening of ICL2 is caused by interactions with ICD.

In summary, we recommend a further revision that places emphasis on the movement of ICL2, which is crucial for clarifying the combined model. Once these revisions are addressed, this work could be considered for publication, as it offers new insights into the activation mechanism of this important plant channel.

Reviewer #2:

Remarks to the Author:

The authors took most of the comments of the referee into account, performed new experiments and changed the ms accordingly.

As suggested the authors mutated the positively charged residues (KRRxxK) of AtSLAC1 and performed mutant analysis of the corresponding residues of phosphor-insensitive KnSLAC1 (EERxxA), MpSLAC1 (VGRxxE) and various other amino acid substitutions in the ICL2 motif. Whole cell patch clamp analysis was used to monitor current densities that strengthen their conclusion that "ICL2 is a putative activation switch that can sense the phosphorylation of SLAC1". Nevertheless, the referee is still convinced that coexpression studies of SLAC1 and SLAC1 kinases are a must for biophysical analysis of AtSLAC1 and the respective mutants thereof as outlined in my first review. Appropriate kinases for SLAC1 activation are well known and combined with the authors' mutants the

electrophysiological analysis and thus the interpretation and confirmation of Cryo-EM results need to perform a proper SLAC1 kinase control.

The authors state that pore mutations such as F450A abolish phosphorylation-dependent activation (Chen et al 2010) suggesting that the mutation causes structural changes differing from that of WT. These structural differences between the pore mutation found by Chen et al 2010 and the open structure reported here is now sufficiently discussed by the authors. The same is true for the putative T513 phospho-site.

The authors determined current densities of HEK293T cells transfected with the SLAC1 constructs and cotransfected with a GFP construct to evaluate the effect of mutations. How do the authors ensure that the expression level of the constructs was similar and did not influence the results? This question is critical: only when the determination of expression levels were performed, the conclusions drawn from the patch clamp results are valid. Supplementary Figure 12 b) and c) show that the WT current densities vary a lot, thus the authors need to have an expression control.

Point-by-point response

We appreciate the constructive comments by the reviewers. Below we provide our responses to the issues raised by the reviewers.

Reviewer #1

In the revised manuscript, the authors have addressed the previous points effectively and have provided additional electrophysiological data that adequately addresses the function of ICL2. Additionally, the authors have included comparisons between the wild-type (WT) in an open state, 8D in a closed state, and 6D in both open and closed states. Furthermore, the authors have identified a new element called the "latch" and have discussed the significance of a stable closed conformation. In order to enhance the interpretation of SLAC1 conformational changes, we propose the following perspectives for further discussion:

1. Line 117: The authors suggest that the significant decrease in current density observed with the S59D or S86D mutations in the 6D mutant may be due to poor natural phosphorylation mimicry. To further support this viewpoint, could co-expression with OST1 be considered based on the 6D mutant?

Co-expression of OST1 would be an attractive way to compare the effects of phosphorylation and aspartate (or glutamate) mutation of S59 and S86. These two sites, S59 and S86, are already known to be phosphorylated by OST1 [Vahisalu et al, Plant J, 2010]. However, these two sites were identified to be highly phosphorylated by endogenous kinase of HEK293 cell and alanine mutations showed impaired current density in previous study [Li et al, Nat Commun, 2022]. OST1-fused AtSLAC1 exhibited much higher current density than AtSLAC1 or co-expression of AtSLAC1 and OST1 in *Xenopus* oocyte whereas OST1-fused AtSLAC1 in HEK293 cell represent non-significant increases of current density compared to solely AtSLAC1 expression [Deng et al, PNAS, 2021; and Li et al, Nat Commun, 2022]. It seems that other expression host is needed to test the co-expression of OST1 as reviewer's suggestion. We thought that these two sites of 6D mutants were also phosphorylated in our study. Both S59D and S86D replaced the phosphorylation and showed similar defective effect (Supplementary Fig. 1e-g). Taken together, we assumed S59D and S86D as phospho-defective mutations.

2. Line 197: The authors introduce the concept of the latch as an additional inhibitory component in higher plants. It would be interesting to investigate its existence and verify it through electrophysiology experiments. For example, deletion or replacement of a specific segment within 105-112 could be explored.

Pursuant to the reviewer's comment, we introduced alanine substitutions for two conserved phenylalanine residues (F106A/F109A) to disrupt conserved hydrophobic interactions of latch toward plug and TMD. We also tried to replace entire eight latch residues into poly-glycine (8G). We hypothesized that these mutations interfere latch binding and showed increased current densities of AtSLAC1 8D, but not in AtSLAC1 6D due to weak latch binding. We observed that F106A/F109A and phosphomimetic mutant of patch 1 (called as "4D", S113D/T114D/S116D/S120D), but not 8G, represented increased current densities compared to AtSLAC1 8D background. However, we also observed increased expression levels of these three mutants (Supplementary Fig. 2c). For this reason, we cannot conclude that current density increases of AtSLAC1 8D are caused by disruption of latch binding. We revised our manuscript to reflect these points as follows:

[Lines 429-468 of the revised manuscript] "Recent studies suggest that AtSLAC1 harbors two major phosphorylation sites S59 and S86, and two patches (patch 1 and 2) that are crowded with minor phosphorylation sites^{28,31}. Our structures uncover that distribution of phosphorylation sites in the intracellular space is restricted by spatial constraints from the ordered plug and latch (Fig. 5a). To confirm whether the restricted spatial distribution of phosphorylation sites, plug and latch of AtSLAC1 are general features of SLAC1, we conducted multiple sequence alignments of orthologous SLAC1 sequences from dicot, monocot and basal plants, and two ancient SLAC1 orthologues (Supplementary Fig. 16). All plant SLAC1s except the two ancient SLAC1 orthologues have the well conserved N- and C-terminal plug (Supplementary Fig. 16). The latch is well conserved in dicot and monocot SLAC1s, and also found in some basal plant SLAC1s (Fig. 5b, Supplementary Fig. 16). In the N-terminal half of the latch sequence motif, aromatic residues F106 and F109 in the latch interact with both the plug and TMD at once, with larger BSAs (121.31 \AA^2 and 179.19 \AA^2) than other residues (Fig. 5b, c). The motif featuring the two aromatic residues (F/Y)xxF is highly conserved, reflecting its importance for binding (Fig. 5b). By contrast, solvent exposed residues D105, S107 and M108 are less conserved in the latch of dicot and basal plant SLAC1s (Fig. 5b, c). Interestingly, the C-terminal half of the latch (¹¹⁰RTK¹¹²) is conserved despite small BSAs (Fig. 5b). We found a conserved pattern that the OST1-mediated

phosphorylation motif ($^{110}\text{RxxS}^{113}$)⁴⁰ spans the C-terminal half of the latch and the first serine (S113) of the patch 1 (Fig. 5a, b, Supplementary Fig. 16).

Multiple sequence alignment uncovered that the latch and the patch 1 motifs are juxtaposed and the OST1-mediated phosphorylation motif overlaps with both the latch and patch 1 motifs (Fig. 5b). Structural analysis revealed that AtSLAC1 8D represented a low conductivity closed state with a clear latch density whereas AtSLAC1 6D seemed to be in a conformational equilibrium between the open and closed states with a weak latch density (Supplementary Fig. 11b). Therefore, we hypothesized that either latch disruption or patch 1 phosphorylation is effective in the activation of AtSLAC1 8D, but not in 6D. To disrupt latch binding, we substituted two conserved phenylalanine residues for alanine (F106A/F109A) or eight residues of the entire latch motif for glycine (8G). To test phosphorylation effects on the patch 1, we introduced the phosphomimetic mutations (S113D/T114D/S116D/S120D), called 4D. F106A/F109A and 4D, but not 8G, in the background of AtSLAC1 8D increased current densities compared to AtSLAC1 8D itself (Fig. 5d). By contrast, none of these three mutants in the background of AtSLAC1 6D showed noticeable increase in the current densities. Since all the latch or patch 1 mutants represented higher expression levels than AtSLAC1 WT, 6D and 8D (Supplementary Fig. 2a, c), we cannot exclude the possibility that the increased current densities by the F106A/F109A and 4D mutants in the background of AtSLAC1 8D are caused by factors other than the inhibition of latch binding. Taken together, multiple sequence alignment, structural and electrophysiological analyses suggest that the plug and the latch are common inhibitory components of SLAC1 in plants. Further studies are necessary to corroborate functional interplay among the latch, patch 1 and phosphorylation.”

3. Line 210: By comparing the consistently extended ICL2 in WT SLAC1 with 6D SLAC1 in the open state, the authors demonstrate that ICL2 is a prominent feature. It would be valuable to improve the resolution of ICL2 in order to confirm the density in this region (Supplementary Fig. 9b, c, d) and determine if there is an interaction between ICD and ICL2.

As reviewer’s suggestion, it would be worth improving resolution of open state structure to uncover structural basis for interactions between ICL2 and ICD. However, it is expected that heterogeneous conformation around ICL2 make it difficult to get ordered structure even if we achieve high resolution structure. Homogeneous phosphorylation of highly potent sites may enable us to get higher resolution

around ICL2. We would be pleased to investigate these things in follow-up studies.

Additionally, the authors could discuss why SLAC1 alone maintains an open conformation in the absence of kinase and whether this conflicts with the activation mechanism in stomata (Geiger D et al 2009; Maierhofer T et al 2014).

We do not think that SLAC1 alone represents an open conformation in both electrophysiology and structural study. Similar to HEK293 cell, unknown endogenous kinases in insect cell may contribute to phosphorylation of SLAC1. We revised our manuscript to provide a clear explanation.

[Lines 139-145 of the revised manuscript] “Bimodal conformational landscape from cryo-EM data is consistent with electrophysiological characterization. Since we prepared AtSLAC1 from an insect cell expression system, AtSLAC1 were likely to be phosphorylated by endogenous kinases of insect cell origin as similarly reported³¹. Highly conductive AtSLAC1 WT showed predominantly the open conformation, whereas less conductive AtSLAC1 7D and 8D exhibited mostly the closed conformation (Fig. 1a, Supplementary Fig. 1, 5, 6). Moderately conductive AtSLAC1 6D mutant adopted both the open and closed conformations (Fig. 1a, Supplementary Fig. 1, 3, 4).”

4. Line 219: The authors conclude that the interaction between ICL2 and ICD is likely to induce conformational changes, but there appears to be a lack of evidence in the structure. Is it possible that the movement of the latch and plug creates space by relieving steric hindrances, thereby allowing ICL2 to move downwards? Instead of focusing solely on the conformational change associated with the release of inhibition, it may be informative to compare the structural differences in ICL2 between WT and NNNLCN to confirm that the loosening of ICL2 is caused by interactions with ICD.

BdSLAC1 structure (PDB: 7EN0) showed ICL2 in bent state without stable plug structure [Deng et al, PNAS, 2021]. Alphafold2 prediction (Q9LD83) also represented ICL2 of AtSLAC1 as bent state. In predicted structure, there's no meaningful interaction between plug and ICL2 due to imperfect prediction of plug region. It suggested that bent state of ICL2 is energetically favorable even though plug is released. We presumed that additional driving force is necessary for ICL2 transition from bent

to extended state. Based on open structure and electrophysiology, the interaction between ICL2 and phosphorylated ICD is the most plausible explanation for conformational changes. We revised our manuscript to explain these points as follows:

[Lines 494-514 of the revised manuscript] “From our cryo-EM structures, we found structural features supporting both the binding-activation and the inhibition-release models. In the closed state structures, the plug consisting of the N- (residues 148-182) and C- (506-517) terminal regions of ICD and the latch (residues 105-112) make contacts with TMD in similar manners despite different phosphorylation states. Phosphorylation sites are located in the disordered region of ICD except S107, S152 and T513 on the plug^{28,29,31} (Supplementary Fig. 11c, 12). In the open state, the plug and latch are released upon phosphorylation on multiple serine/threonine residues in the disordered region of ICD. However, in data processing of AtSLAC1 6D, we observed populations with closed-like structures representing a plug-free closed TMD, reminiscent of the closed BdSLAC1 structure (Supplementary Fig. 7). AlphaFold2 prediction of AtSLAC1 (<https://alphafold.ebi.ac.uk/entry/Q9LD83>) also apparently features a closed TMD with a partially unfolded plug. The ICL2 was predicted to maintain a bent state despite a negligible interaction toward the plug^{41,42}. These results suggest that phosphorylation events, presumably inducing destabilization and subsequent dissociation of the plug and latch regions, are necessary for SLAC1 to escape from inhibition, reflecting the inhibition-release model, but not sufficient to convert it into the open state. In our open structures, phosphorylation events are connected to the channel opening via interactions between the phosphorylated ICD and ICL2, a feature of the binding-activation model (Fig. 1). The negative charges imposed by the phosphate groups attract the cluster of positively charged residues in ICL2, with concomitant unwinding of TM5, bending of TM7 and following intracellular rearrangements rendering the reshaping into inverted Y-shaped pore and widening of pore (Fig. 2, 3).”

Reviewer #2

The authors took most of the comments of the referee into account, performed new experiments and changed the ms accordingly. As suggested the authors mutated the positively charged residues (KRRxxK) of AtSLAC1 and performed mutant analysis of the corresponding residues of phosphor-insensitive KnSLAC1 (EERxxA), MpSLAC1 (VGRxxE) and various other amino

acid substitutions in the ICL2 motif. Whole cell patch clamp analysis was used to monitor current densities that strengthen their conclusion that “ICL2 is a putative activation switch that can sense the phosphorylation of SLAC1”. Nevertheless, the referee is still convinced that coexpression expression studies of SLAC1 and SLAC1 kinases are a must for biophysical analysis of AtSLAC1 and the respective mutants thereof as outlined in my first review. Appropriate kinases for SLAC1 activation are well known and combined with the authors’ mutants the electrophysiological analysis and thus the interpretation and confirmation of Cryo-EM results need to perform a proper SLAC1 kinase control.

Pursuant to the reviewer’s concern, we added a paragraph that explains limitations of our study about potential phosphorylation by unknown kinases of heterologous origins.

[Lines 482-492 of the revised manuscript] “Our structural and electrophysiological results implicate that AtSLAC1 constructs were likely to be phosphorylated by unknown endogenous kinases of insect and human cells as previously reported³¹. Such phosphorylation by kinases from heterologous origins may have led to heterogeneity in phosphorylation states, which in turn would complicate the interpretation of phosphorylation effects by phosphomimetic mutants. We also observed that some aspartate or glutamate mutations were phospho-defective (Supplementary Fig. 1f). These observations could impose uncertainty in the interpretation of the effect of each prospective phosphorylation site. Further studies would be needed under precise control of phosphorylation state to unequivocally confirm the effects of phosphorylation on SLAC1. Nonetheless, our cryo-EM structures combined with electrophysiological data enabled us to unveil the important structural features for SLAC1 activation.”

The authors state that pore mutations such as F450A abolish phosphorylation-dependent activation (Chen et al 2010) suggesting that the mutation causes structural changes differing from that of WT. These structural differences between the pore mutation found by Chen et al 2010 and the open structure reported here is now sufficiently discussed by the authors. The same is true for the putative T513 phospho-site. The authors determined current densities of HEK293T cells transfected with the SLAC1 constructs and cotransfected with a GFP construct

to evaluate the effect of mutations. How do the authors ensure that the expression level of the constructs was similar and did not influence the results? This question is critical: only when the determination of expression levels was performed, the conclusions drawn from the patch clamp results are valid. Supplementary Figure 12 b) and c) show that the WT current densities vary a lot, thus the authors need to have an expression control.

We deeply appreciate the reviewer's suggestion on checking expression levels of mutants used for electrophysiological experiments. We tested expression levels of constructs used in this study (Supplementary Fig. 2). Briefly, biotinylated HA-tagged AtSLAC1 WT and mutants on cell surface were pulled down by an avidin resin and detected by immunoblotting using an anti-HA antibody. Most constructs showed similar surface localization ratio R_{surface} : $R_{\text{surface}} = \frac{(I_{\text{surface}}/I_{\text{TfR}})}{(I_{\text{total}}/I_{\text{actin}})}$ where I_{surface} , I_{TfR} , I_{total} , and I_{actin} refer to band intensities of surface proteins, transferrin receptor, total proteins, and actin, respectively. These results indicate that mutations in this study did not cause either mis-folding or mis-localization. Expression levels of phosphomimetic mutants used in the cryo-EM analysis did not differ from that of AtSLAC1 WT (Supplementary Fig. 2a). Notably, we observed increased expression levels of some ICL2 mutants. Since these ICL2 mutants showed decreased current density despite their increased expression levels (Fig. 1g, h, Supplementary Fig. 2b), there is no issue in interpretation of the electrophysiological data. We prepared additional mutants in which latch (F106A/F109A or 8G) and patch 1 (4D, S113D/T114D/S116D/S120D) regions were disrupted to investigate latch binding and its regulation by phosphorylation. We hypothesized that disruption of latch binding by these mutants might activate AtSLAC1 8D, but not 6D. The F106A/F109A and 4D mutants in the AtSLAC1 8D background indeed exhibited increased current densities compared to those by AtSLAC1 8D itself (Fig. 5d). By contrast, these latch or patch 1 mutants in the AtSLAC1 6D background did not exhibit any difference in current densities from those by AtSLAC1 6D itself (Fig. 5d). Since all the latch or patch 1 mutants showed significant increase in their expression levels (Supplementary Fig. 2c), we cannot exclude the possibility that these increased expression levels or other factors might have affected the increased current densities of F106A/F109A and 4D mutants in the AtSLAC1 8D background. We would be pleased to investigate functional interplay between these components in follow-up studies. We revised our manuscript to reflect these points as follows:

[Lines 121 of the revised manuscript] "These mutations did not affect expression level and

localization (Supplementary Fig. 2a).”

[Lines 240-241 of the revised manuscript] “ICL2 mutants showed similar or higher expression level than that of WT AtSLAC1 (Supplementary Fig. 2b).”

[Lines 454-465 of the revised manuscript] “Therefore, we hypothesized that either latch disruption or patch 1 phosphorylation is effective in the activation of AtSLAC1 8D, but not in 6D. To disrupt latch binding, we substituted two conserved phenylalanine residues for alanine (F106A/F109A) or eight residues of the entire latch motif for glycine (8G). To test phosphorylation effects on the patch 1, we introduced the phosphomimetic mutations (S113D/T114D/S116D/S120D), called 4D. F106A/F109A and 4D, but not 8G, in the background of AtSLAC1 8D increased current densities compared to AtSLAC1 8D itself (Fig. 5d). By contrast, none of these three mutants in the background of AtSLAC1 6D showed noticeable increase in the current densities. Since all the latch or patch 1 mutants represented higher expression levels than AtSLAC1 WT, 6D and 8D (Supplementary Fig. 2a, c), we cannot exclude the possibility that the increased current densities by the F106A/F109A and 4D mutants in the background of AtSLAC1 8D are caused by factors other than the inhibition of latch binding.”

[Lines 751-766 of the revised manuscript] “To test cell surface expression level of AtSLAC1, surface biotinylation was performed. HEK293T cells were transfected with wild-type and mutant constructs using the Transporter 5™ Transfection Reagent and incubated for 40 hours. Cells were washed with ice-cold PBS and biotinylated using 2 mL of 0.25 mg/mL (per 35 mm) EZ-Link Sulfo-NHS-SS-Biotin for 15 min on ice. Remaining reactive biotin was washed out with PBS and additionally quenched with 50 mM glycine pH 7.5. Cells were harvested and lysed in a lysis buffer (1% Triton X-100, 1X PBS, 1X Protease inhibitor cocktail). Biotinylated plasma membrane (surface) proteins were purified from whole cell lysate (total) using NeutrAvidin Plus UltraLink resin. After washing the resin with the lysis buffer, proteins were eluted from resin by 2X LDS sample buffer. Total and surface proteins were separated on a Bolt™ 4-12% Bis-Tris Plus Gel (Invitrogen) and transferred to a PVDF membrane using iBlot 2 Transfer Stack. 250 - 500 ng antibodies [anti-HA (ab9110, Abcam), anti-Actin (8457S, Cell Signaling Technology), and anti-transferrin receptor (TfR) (13208S, Cell Signaling Technology)]/mL iBind Flex Solution were used for immunoblotting. Protein bands were

visualized by Clarity Western ECL Substrate and images were acquired by the ChemiDOC imaging system. Three independent experiments were conducted. Densitometric analysis was performed by ImageJ⁵² software.”

Reviewers' Comments:

Reviewer #1:

Remarks to the Author:

In this revision, the authors have conducted a more comprehensive investigation of the latch and ICL2. They have enriched their study with additional electrophysiological experiments and provided a meticulous comparison of their findings with previous research, effectively addressing the concerns raised by the referees. The "combined model" proposed in this work can be considered for publication, as it has the potential to expand the interpretation and discussion within the field of SLAC1.

Reviewer #2:

Remarks to the Author:

My vote for the paper reject and submit "major revision".

Why I reached this decision?

The authors did not perform the "must do experiments" of expressing SLAC1 in oocytes.

In contrast to HEK cells Xenopus oocytes do not phosphorylate SLAC1.

What we know about the regulation of SLAC1 via phosphorylation basically comes from oocyte experiments.

Expressing SLAC1 in oocytes and performing voltage-clamp experiments every lab in the word can do within a few weeks.

If the author resist to perform the experiment requested, they do not want to confirm or falsify their model.

Without the "must do experiments" the paper should not published in Nature Communications.

Point-by-point response

We appreciate the constructive comments by the reviewers. Below we provide our responses to the issues raised by the reviewers.

Reviewer #1

In this revision, the authors have conducted a more comprehensive investigation of the latch and ICL2. They have enriched their study with additional electrophysiological experiments and provided a meticulous comparison of their findings with previous research, effectively addressing the concerns raised by the referees. The "combined model" proposed in this work can be considered for publication, as it has the potential to expand the interpretation and discussion within the field of SLAC1.

We appreciated the reviewer's encouragement. We will make every effort to refine our manuscript to properly provide a new perspective in the SLAC1 field.

Reviewer #2

*My vote for the paper: reject and submit "major revision". Why I reached this decision? The authors did not perform the "must do experiments" of expressing SLAC1 in oocytes. In contrast to HEK cells *Xenopus* oocytes do not phosphorylate SLAC1. What we know about the regulation of SLAC1 via phosphorylation basically comes from oocyte experiments. Expressing SLAC1 in oocytes and performing voltage-clamp experiments every lab in the word can do within a few weeks. If the authors resist to perform the experiment requested, they do not want to confirm or falsify their model. Without the "must do experiments" the paper should not be published in *Nature Communications*.*

We apologize for being unable to resolve the issue of performing electrophysiology experiments in *Xenopus* oocytes to the satisfaction of the reviewer. To address the reviewer's concern, we revised the Discussion section to point out the advantages of electrophysiological characterization in *Xenopus* oocytes expression system and to note limitations of our study in that unknown phosphorylation effects by endogenous kinases in HEK cells could introduce uncertainties in the interpretation of our experimental results.

[Lines 484-497 of the revised manuscript; added sentences are highlighted as brown.] “Our structural and electrophysiological results implicate that AtSLAC1 constructs were likely to be phosphorylated by unknown endogenous kinases of insect and human cells as previously reported³¹. Such phosphorylation by kinases from heterologous origins may have led to heterogeneity in phosphorylation states, which in turn would complicate the interpretation of phosphorylation effects by phosphomimetic mutants. We also observed that some aspartate or glutamate mutations were phospho-defective (Supplementary Fig. 1f). These observations could impose uncertainty in the interpretation of the effect of each prospective phosphorylation site. In *Xenopus* oocyte expression system, SLAC1 alone is electrically silent and activated only by co-expression of cognate kinases^{20,21,41}. Electrophysiological characterization of SLAC1 mutants with cognate kinases in oocytes might be better suited to disentangle subtle and complicated phosphorylation effects. Further studies would be needed under precise control of phosphorylation state to unequivocally confirm the effects of phosphorylation on SLAC1. Nonetheless, our cryo-EM structures combined with electrophysiological data enabled us to unveil the important structural features for SLAC1 activation.”